# HLD: APPROXIMATE HIERARCHICAL LINGUISTIC DISTRIBUTION MODELING FOR LLM-GENERATED TEXT DETECTION

**Rui Guo**[1,2][*]**, Weibin Zeng**[1][*]**, Fuzhang Wu**[1][†]**, Yan Kong**[1]**, Sicheng Shen**[1]**,
Yanjun Wu**[1]**, Weiming Dong**[3]

[1]Institute of Software, Chinese Academy of Sciences
[2]Zhengzhou University
[3]Institute of Automation, Chinese Academy of Sciences
`guorui2002@gs.zzu.edu.cn`, `{zengweibin, fuzhang}@iscas.ac.cn`

## ABSTRACT

The widespread deployment of large language models (LLMs) has made the reliable detection of AI-generated text a crucial task. However, existing zero-shot detectors typically rely on proxy models to approximate probability distributions of unknown source models at a single token level. Such approaches limit detection effectiveness and make the results highly sensitive to the choice of proxy models. In contrast, supervised classifiers are often detected as black boxes, sacrificing interpretability in the detection process. To address these limitations, we propose a novel detection framework that identifies LLM-generated text by approximating **H**ierarchical **L**inguistic **D**istributions–**HLD-Detector**[1]. Specifically, we leverage n-grams to capture the feature distribution of human-written and machine-generated text across the word, syntactic, and semantic levels, and perform LLM-generated text detection by comparing these distributions under the Bayesian theory. By progressively modeling the linguistic distribution from shallow-level (token/word), then medium-level (syntactic), and ultimately high-level (semantic representations), our method mitigates the shortcomings of previous single feature level detection, improving both robustness and overall performance. Additionally, HLD-Detector requires only a small amount of offline corpus for distribution estimation, instead of relying on online approximation with large proxy models, resulting in significantly lower computational overhead. Extensive experiments have verified the superiority of our method in detection tasks such as multi-llm and multi-domain scenarios, achieving the current SOTA performance.

## 1 INTRODUCTION

Large language models (LLMs) have achieved remarkable breakthroughs in recent years (Guo et al., 2025; Achiam et al., 2023). Their powerful text generation capabilities are transforming content creation, yet they also introduce serious risks: automatically fabricated news (Hu et al., 2025), academic misconduct (Perkins, 2023), and other forms of misuse threaten the stability of social trust systems and information ecosystems (Weidinger et al., 2021; Lee et al., 2024). Against this backdrop, developing accurate, efficient, and reliable AI-generated text detection technologies has become a critical and urgent task (Wu et al., 2025; Abdali et al., 2024).

To address this challenge, the community has explored diverse detection methods (as illustrated in Fig. 1). An effective approach involves fine-tuning pretrained models for detection (Abassy et al., 2024; Bahad et al., 2024; Hee Lee & Jang, 2024). Under this paradigm, input text is transformed to high-dimensional neural representations and subjected to classification. While this strategy can

---

[*]Equal contribution
[†]Corresponding author
[1]We release our code at `https://github.com/nefugr/HLD-Detector`

achieve strong performance on specific datasets, it fundamentally relies on learning neural representations to fit the training data distribution. Therefore, when the test data distribution deviates from the training data, detection performance will deteriorate significantly. Moreover, as the detector's decisions entirely depend on the model's internal hidden states, which are difficult to interpret as concrete features, the resulting predictions offer limited interpretability.

Another line of research focuses on zero-shot detection (Mitchell et al., 2023; Bao et al., 2023; Hans et al., 2024; Bao et al., 2025), which exploits the tendency of LLMs to prefer high-probability tokens over the greater variability of human writing. While interpretable, this framework depends on shallow token-level distributions that are easily disrupted by synonym perturbations, limiting the robustness of the detector. In addition, it also typically relies on surrogate models to approximate the token probability distribution of the source model. However, the popular commercial models such as GPT (Achiam et al., 2023) and Gemini (Team et al., 2023) are black-box, making it difficult for proxy models to capture their true distributions, which in turn degrade the detection performance. Moreover, surrogate inference is often computationally costly, adding significant latency.

In our view, LLMs exhibit remarkable text generation capabilities, producing outputs that are highly similar to human. As a result, distinguishing between machine-generated text (MGT) and human-written text (HWT) based on a single feature is challenging. Prior work indicates that MGT exhibits characteristic patterns in word choice and syntactic structure (Chawla, 2024; Durward & Thomson, 2024); Wang et al. (2025) further report LLMs have more rigid semantic behavior, suggesting systematic differences in semantic representations. These observations motivate us to move beyond any single feature hierarchy (such as token probabilities or neural embeddings alone) and develop a hierarchical modeling of linguistic distributions.

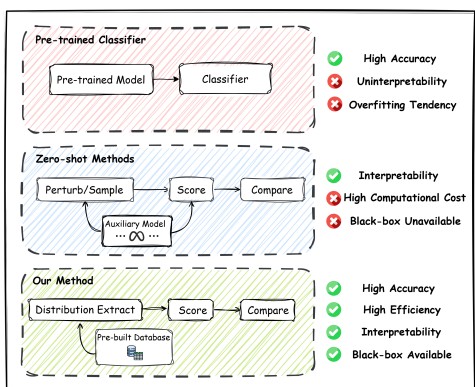

Figure 1: Comparison of different methods for AI-generated text detection.

Motivated by this insight, we propose a novel LLM-generated text detection framework (**HLD-Detector**) that captures the differences between hierarchical linguistic distributions of MGT and HWT. Specifically, HLD progressively models the feature distributions from shallow word level to medium syntactic level (including part-of-speech and dependency) and ultimately to high semantic level. At the word level, it captures the basic distribution differences to achieve effective classification. At the syntactic level, it reveals more general distributional patterns through deeper structures, strengthening the detector's generalization ability. Finally, at the highest semantic level, it encodes content into embedding representations, and compare the distribution differences in representations to further enhance detection robustness under adversarial scenarios. As shown in Fig. 2, the hierarchical linguistic features demonstrate a clear distributional divergence between the two sources, validating the efficacy of our HLD design.

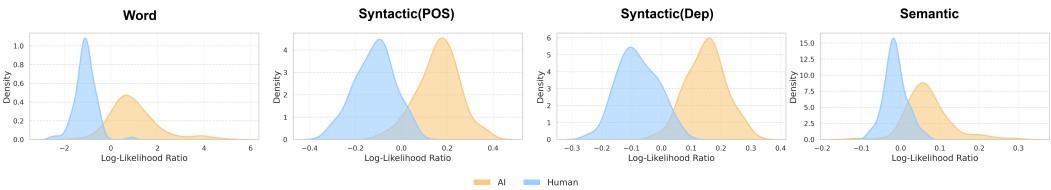

Figure 2: Density distributions of the Hierarchical Linguistic feature Log-Likelihood Ratios ($z_{word}$, $z_{pos}$, $z_{dep}$, $z_{semantic}$) for AI-generated (yellow) and Human-written (blue) text.

To estimate hierarchical linguistic feature distributions between MGT and HWT, we employ an n-gram model (Shannon, 1948), which aims to avoid the challenges of aligning distributions from black-box generators and allow efficient inference. Estimating the full probability distribution for ei-

ther source is computationally intractable, as required data grows exponentially with content length. Focusing on distributional differences, we assume a Markovian property and truncate the context to length $k$. Although this approximation does not capture the full-context distribution, it suffices for comparing the distributions of the two sources, and can be reliably estimated from only a few samples.

Eventually, supported by Bayesian theory, HLD calculates the log-likelihood ratios between MGT and HWT across the above hierarchical linguistic distributions, obtained via n-grams and inputs these ratios into a classifier for AI-generated content detection. In summary, our contributions are as follows:

1. We propose **HLD-Detector**, a framework that systematically models the hierarchical linguistic distribution differences between MGT and HWT. By progressively capturing features from shallow lexical, to medium syntactic, and high-level semantic representations, our method substantially improves detection performance, generalization, and robustness.

2. To avoid distribution alignment challenges from black-box models and the high cost of proxy-model-based inference, we leverage n-grams with a Markov assumption to estimate MGT and HWT distributions from limited samples and significantly speeding up inference.

3. We conduct comprehensive AI-generated text detection tasks on the DetectRL (Wu et al., 2024) dataset. Our HLD significantly outperforms existing baselines, demonstrating the effectiveness of hierarchical linguistic distribution modeling and n-gram based distribution estimation.

## 2 METHOD

**Task Definition.** This task aims to determine whether a given text is machine-generated (MGT) or human-written (HWT), which we formulate as a binary classification problem. Our approach is founded on the observation that LLMs generate text via maximum likelihood sampling, their outputs follow distinct distributions from HWT across hierarchical linguistic features, including word, syntactic, and semantic aspects(Muñoz-Ortiz et al., 2024). Motivated by this observation, HLD is designed to characterize input text through a diverse set of hierarchical linguistic features, and modeling the distributional discrepancies between HWT and MGT along these dimensions to achieve reliable classification. Formally, let $X = (x_1, x_2, \ldots, x_n)$ denotes a text sequence of length $n$, $Y \in \{0, 1\}$ be the class label (HWT vs. MGT). We first transform the $X$ into feature sequences at different linguistic levels, formulated as

$$F_j = \phi_j(X) = (f_0, f_1, f_2, \ldots, f_n), \tag{1}$$

where $\{\phi_j(\cdot)\}_{j=1}^4$ denotes a linguistic transformation function, corresponding respectively to word-level, syntactic-level (part-of-speech, dependency), and semantic-level.

Then, for each linguistic sequence $F_j$ ($\forall j \in [4]$), the classification rule is defined as:

$$\hat{Y} = \begin{cases} 1, & \text{if } \frac{P(Y=1|F_j)}{P(Y=0|F_j)} \geq \frac{\tau}{1-\tau}, \\ 0, & \text{otherwise.} \end{cases}$$

where $\tau \in (0, 1)$ is decision threshold. By Bayes' theorem and the chain rule, the likelihood ratio can be decomposed into conditional probability ratio over the feature tokens $f_i$ under the distribution of different classes $Y$.

$$\frac{P(Y=1 \mid F_j)}{P(Y=0 \mid F_j)} = \frac{P(F_j \mid Y=1)P(Y=1)}{P(F_j \mid Y=0)P(Y=0)} \propto \frac{\prod_{i=1}^n P(f_i \mid Y=1, f_1, \ldots, f_{i-1})}{\prod_{i=1}^n P(f_i \mid Y=0, f_1, \ldots, f_{i-1})}, \tag{2}$$

**Markov Approximation.** Equation 2 indicates that LLM-generated text can be identified by comparing the likelihood ratio (LR) of feature distributions between MGT and HWT. Instead of relying on surrogate models to acquire the distributions typically employed by zero-shot detectors (Mitchell et al., 2023; Bao et al., 2023), we derive them directly from the text samples through n-gram statistics (Shannon, 1948). This reduces the complexity of aligning distributions from black-box LLMs

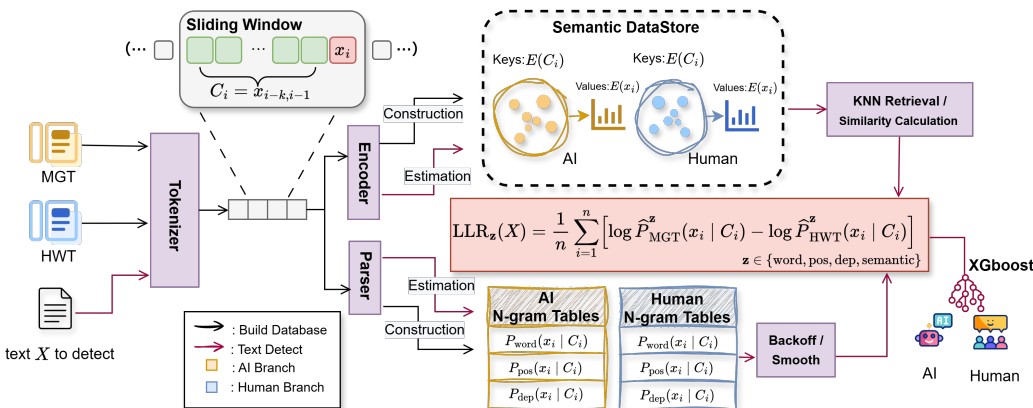

Figure 3: Architecture of our HLD method. The input text is segmented with a sliding window, and for each context–token pair, a log-likelihood ratio (LLR) is computed by contrasting probabilities under AI- and human-authored databases. This comparison is performed at hierarchical levels: word, syntactic, and semantic. Finally, the resulting LLRs are aggregated by an XGBoost classifier for the final decision.

and supporting efficient inference. However, evaluating $P(f_i \mid Y, f_{<i})$ over the full context is computationally expensive for n-grams. Based on the Markov chain assumption, distant linguistic tokens $(f_1, \ldots, f_{i-k-1})$ are expected to have limited influence on $f_i$. Therefore, we approximate

$$\frac{P(f_i \mid Y = 1, f_{<i})}{P(f_i \mid Y = 0, f_{<i})} \approx \frac{P(f_i \mid Y = 1, f_{i-k:i-1})}{P(f_i \mid Y = 0, f_{i-k:i-1})},$$

which allows n-grams to be learned more easily via only a small amount of training data. Furthermore, to keep numerical stability and eliminate length effects of LR, we take the logarithm and normalize by sequence length $n$, and the decision rule becomes:

$$\hat{Y} = \mathbb{I}\left( \frac{1}{n} \sum_{i=1}^{n} \log \frac{\hat{P}_{\text{HWT}}(f_i \mid f_{i-k:i-1})}{\hat{P}_{\text{MGT}}(f_i \mid f_{i-k:i-1})} = \mathbf{z} \geq \epsilon \right), \tag{3}$$

where $\mathbb{I}(\cdot)$ is the indicator function, $\mathbf{z}$ denotes the log likelihood ratio and $\epsilon$ is the decision threshold.

## 2.1 HIERARCHICAL LINGUISTIC DISTRIBUTION MODELING

In this subsection, we present the modeling process of HLD across hierarchical linguistic feature levels. Fig. 3 shows the architecture of HLD.

**Word Distribution Modeling.** Fast-DetectGPT (Bao et al., 2023) posits that human and LLM differs in word selection based on preceding context and it uses a conditional probability curvature to estimate this discrepancy. Motivated by this idea, we begin by distinguishing MGT and HWT through modeling the word-level distribution differences of the two sources. Specifically, we first transform the samples via $\phi_{\text{word}}$ into word sequences $F_{\text{word}} = \phi_{\text{word}}(X)$. We then tokenize these words and build n-gram language models for both MGT and HWT sources. In detection phase, HLD leverages these distributions and applies smoothing and back-off strategies to quantify the discrepancy between $\hat{P}_{\text{MGT}}^{\text{word}}(f_i \mid f_{i-k:i-1})$ and $\hat{P}_{\text{HWT}}^{\text{word}}(f_i \mid f_{i-k:i-1})$ for the evaluated text:

$$\hat{P}_Y(f_i \mid f_{i-k:i-1}) = \begin{cases} \frac{C_Y(f_{i-k:i-1}, f_i) + \delta}{C_Y(f_{i-k:i-1}) + \delta \cdot |V|}, & \text{if } C_Y(f_{i-k:i-1}) > 0, \\ \hat{P}_Y(f_i \mid f_{i-k+1:i-1}), & \text{otherwise}, \end{cases} \tag{4}$$

where $C_Y(f_{i-k:i-1}, f_i)$ denotes the occurrence that sequence $f_{i-k:i-1}$ is followed by $f_i$ under word distribution of $Y$, $C_Y(f_{i-k:i-1})$ denotes the total occurrences of $f_{i-k:i-1}$ in $Y$'s word-level distribution, $\delta$ is the smoothing factor (set to $1 \times 10^{-6}$), and $|V|$ represents the vocabulary size. The resulting

word-level conditional probabilities are subsequently converted into the log ratio $\mathbf{z}_{\text{word}}$ according to Eq.3 for the final classification.

**Syntactic Distribution Modeling.** As prior work reveals that AI-generated text exhibits characteristic patterns at deeper syntactic levels (Fröhling & Zubiaga, 2021), we further incorporate syntactic features to capture more general structural regularities and improve the generalization of HLD. We map the training sequences $X$ into part-of-speech (POS) tag sequences $F_{\text{POS}} = \phi_{\text{pos}}(X)$ [2] and dependency (Dep) relation sequences $F_{\text{Dep}} = \phi_{\text{Dep}}(X)$ [3]. We then model the distributions over these "POS" and "Dep" sequences analogously to the word distribution, using separate n-gram models for each feature type. During inference, we estimate the conditional probabilities $\hat{P}_Y^{\text{POS}}(f_i \mid f_{i-k:i-1})$ and $\hat{P}_Y^{\text{Dep}}(f_i \mid f_{i-k:i-1})$ under the respective syntactic distributions for an input text, and derive the corresponding distributional differences $\mathbf{z}_{\text{pos}}$ and $\mathbf{z}_{\text{Dep}}$ at syntactic-level.

**Semantic Distribution Modeling.** Considering the vulnerability of word- and syntactic-level features to adversarial paraphrasing, we draw inspiration from Dipper (Krishna et al., 2023), which constructs a semantic retrieval database for attack defense. Accordingly, We transcend token sequences and formalize the conditional probability distribution $P(f_i \mid f_{i-k:i-1})$ in the continuous semantic space. A direct parametric estimation of this distribution $P(\cdot|f_{i-k:i-1})$ is intractable due to the high dimensionality. Inspired by Kernel Density Estimation (KDE)Terrell & Scott (1992), we propose a non-parametric approach to estimate these probabilities. Specifically, we first transform text samples $X$ into sequences of context-target embedding pairs via a pre-trained encoder, denoted as $F_{\text{semantic}} = \phi_{\text{semantic}}(X)$. Each pair consists of a contextual embedding $f_{i-k:i-1}$ and a corresponding target embedding $f_i$. The database $D_Y^{\text{semantic}}$ is constructed offline by processing a corpus for each class Y and storing the resulting context-target embedding pairs. In the detection phase, given a query pair $(f_{i-k:i-1}, f_i)$, we retrieve $M$ nearest context neighbors $\{f_{i-k:i-1,m,Y}\}$ and their associated average target embeddings $\{f_{i,m,Y}^{\text{avg}}\}$ from the database $D_Y^{\text{semantic}}$. The conditional probability is estimated via probabilistic interpolation over these neighbors, following the law of total probability:

$$\hat{P}_Y^{\text{semantic}}(f_i \mid f_{i-k:i-1}) = \sum_{m=1}^{M} \hat{P}(f_i \mid f_{i-k:i-1,m,Y}) \cdot \hat{P}(m \mid f_{i-k:i-1}) \tag{5}$$

In this formulation, both the neighbor weight distribution $\hat{P}(m \mid f_{i-k:i-1})$ and the conditional target probability $\hat{P}(f_i \mid f_{i-k:i-1,m,Y})$ are distributions estimated from cosine similarities using softmax kernels. For instance, the neighbor weight distribution is calculated as:

$$\hat{P}(m \mid f_{i-k:i-1}) = \frac{\exp(\text{sim}(f_{i-k:i-1}, f_{i-k:i-1,m,Y})/\tau_{\text{ctx}})}{\sum_{j=1}^{M} \exp(\text{sim}(f_{i-k:i-1}, f_{i-k:i-1,j,Y})/\tau_{\text{ctx}})} \tag{6}$$

where $\text{sim}(\cdot, \cdot)$ denotes the cosine similarity and $\tau_{\text{ctx}}$ is a temperature parameter (set to 0.1). The conditional target probability is computed analogously, based on the cosine similarity between the query target embedding $f_i$ and the retrieved average target embedding $f_{i,m,Y}^{\text{avg}}$ for each neighbor.

## 2.2 Classification via XGBoost

To fully utilize the hierarchical linguistic distribution differences of MGT and HWT, we additionally train an XGBoost Chen & Guestrin (2016) classifier on these differences. Let $\mathbf{Z} = [\mathbf{z}_{\text{lexical}}, \mathbf{z}_{\text{pos}}, \mathbf{z}_{\text{Dep}}, \mathbf{z}_{\text{semantic}}]^\top \in \mathbb{R}^4$ denote the differences, the classifier $f_\theta(\mathbf{Z}) = \sum_{s=1}^{S} T_s(\mathbf{Z})$ is then learned using XGBoost:

$$\hat{P}(Y = 1 \mid \mathbf{Z}) = \sigma(f_\theta(\mathbf{Z})), \quad \theta^\star = \arg\min_\theta \sum \mathcal{L}(Y, \hat{P}(Y \mid \mathbf{Z})) + \Omega(\theta),$$

where $\sigma(\cdot)$ is the logistic function, $T_s$ are regression trees, $\mathcal{L}$ is the binary cross-entropy loss, and $\Omega$ regularizes tree complexity.

---

[2]See details at `https://spacy.io/usage/linguistic-features#pos-tagging`
[3]See details at `https://spacy.io/usage/linguistic-features#dependency-parse`

Table 1: Performance comparison of all detectors in multi-LLM and multi-domain assessment, with results reported as AUROC (%). The best and second-best scores in each column are highlighted in **bold** and underlined, respectively. * denote the zero-shot methods. Results for other metric are provided in Appendix B.1.

| Detector ↓ | Multi-LLM | | | | | Multi-Domain | | | | |
|---|---|---|---|---|---|---|---|---|---|---|
| | GPT-3.5 | Claude | PaLM-2 | Llama-2 | Avg. | Arxiv | XSum | Writing | Review | Avg. |
| LRR* | 61.61 | 43.30 | 71.17 | 83.65 | 64.93 | 70.54 | 50.09 | 64.65 | 76.61 | 65.47 |
| DetectGPT* | 43.46 | 32.86 | 26.72 | 36.71 | 34.94 | 22.15 | 12.21 | 58.95 | 44.43 | 34.44 |
| Binoculars* | 88.14 | 55.15 | 93.30 | 96.64 | 83.31 | 84.03 | 77.39 | 94.38 | 90.00 | 86.45 |
| Fast-DetectGPT* | 65.56 | 30.01 | 65.99 | 76.79 | 59.59 | 43.69 | 39.19 | 74.21 | 77.02 | 58.53 |
| DNA-GPT* | 61.87 | 48.88 | 71.48 | 75.22 | 64.36 | 55.85 | 72.18 | 66.27 | 69.84 | 66.04 |
| Lastde++* | 69.21 | 41.51 | 73.75 | 80.18 | 66.16 | 67.07 | 57.73 | 69.32 | 72.40 | 66.63 |
| RADAR | 94.58 | 81.15 | 95.10 | 96.80 | 91.91 | 95.04 | **99.78** | 79.34 | 89.63 | 90.95 |
| Ghostbuster | 91.81 | 84.99 | 81.77 | 89.57 | 87.04 | 91.74 | 93.34 | 83.88 | 86.06 | 88.76 |
| RAIDAR | 86.58 | 87.71 | 90.43 | 89.19 | 88.48 | 94.94 | 96.11 | 85.45 | 93.94 | 92.61 |
| DPIC | 99.39 | 93.99 | 95.53 | 98.08 | 96.75 | 99.02 | 98.81 | 94.69 | 97.65 | 97.54 |
| Roberta-base | 99.47 | 98.52 | 96.61 | 98.36 | 98.24 | **100.00** | 99.75 | 96.94 | 99.05 | 98.94 |
| **HLD(Ours)** | **99.74** | **99.48** | **97.87** | **99.38** | **99.12** | 99.45 | 99.41 | **99.68** | **99.84** | **99.60** |

## 3 EXPERIMENTS

### 3.1 SETTINGS

**Benchmark and Evaluation.** To ensure a comprehensive and fair evaluation, we conducted the experiments on DetectRL benchmark (Wu et al., 2024). DetectRL is specifically designed for evaluating LLM-generated text detectors in real-world scenarios, featuring diverse high-risk domains (e.g., arXiv abstracts, XSum news), mainstream LLMs (e.g., GPT-3.5-turbo, Claude), and well-designed adversarial attacks (e.g., paraphrase, perturbation). This makes it an ideal platform for assessing the overall capabilities of our method. We strictly adhere to its established evaluation protocol. In each domain, we evaluate on a predefined balanced test set of 1,000 human-written and 1,000 machine-generated texts. For supervised training, we construct a balanced training set from the DetectRL training corpus, consisting of 1,800 human-written and 1,800 AI-generated texts. We adopt AUROC and F1-score as our evaluation metrics. AUROC measures the overall ranking capability independent of a specific threshold, while the F1-score reflects the practical balance between precision and recall.

**Baselines.** For a extensive evaluation, we compare our method against two categories of representative baselines: (i) Zero-shot Methods: including LRR (Su et al., 2023), DetectGPT (Mitchell et al., 2023), Binoculars (Hans et al., 2024), Fast-DetectGPT (Bao et al., 2023), DNA-GPT (Yang et al., 2024), and Lastde++ (Xu et al., 2025); (ii) Supervised Methods: including RADAR (Hu et al., 2023), GhostBuster (Verma et al., 2024), RAIDAR (Mao et al., 2024), DPIC (Yu et al., 2024), and RoBERTa Classifier (Liu et al., 2019). More details about the baselines are provided in the Appendix A.2.

### 3.2 MAIN RESULT

**Overall Performance.** As shown in Table 1, our proposed HLD achieves state-of-the-art performance on both multi-LLM and multi-domain detection tasks. In the multi-LLM settings, HLD obtains an average AUROC of 99.12%, surpassing the second-best method (RoBERTa-base) by an absolute margin of 0.88 percentage points. In contrast, zero-shot methods achieve an average AUROC around 60% and exhibit significant fluctuations across different source models, exemplified by DetectGPT's poor performance on PaLM-2 (26.72% AUROC). Our method avoids these inconsistencies by directly analyzing the output text, leading to more reliable results across different generators. In the multi-domain evaluation, HLD remains either the best or very close to the best in all domains. While RoBERTa-base shows a slight advantage on "Arxiv" and "Xsum", HLD achieves the highest average AUROC of 99.60%, outperforming all baselines overall.

**Generalization Analysis.** Table 2 reports results under two transfer settings. In the cross-domain setting (trained on Arxiv and tested on XSum, Writing, and Review), all detectors are affected by

Table 2: Generalization results of detectors. The left block reports cross-domain generalization, and the right block reports cross-model generalization. Best and second-best results in each column are marked in **bold** and underlined. * denote the zero-shot methods. Complete results across all domains and models are provided in Table 9 and Table 10.

| Detector↓ Eval→ | Train on Arxiv | | | | Train on GPT-3.5 | | | |
|---|---|---|---|---|---|---|---|---|
| | XSum | Writing | Review | **Avg.** | Claude | PaLM-2 | Llama-2 | **Avg.** |
| LRR* | 40.88 | 38.44 | 55.81 | 45.04 | 24.70 | 61.79 | 75.34 | 53.94 |
| Fast-DetectGPT* | 23.71 | 59.67 | 60.17 | 47.85 | 12.96 | 59.56 | 69.93 | 47.48 |
| GhostBuster | 85.81 | 75.18 | 69.70 | 76.90 | 61.83 | 77.43 | 83.66 | 74.31 |
| RAIDAR | 87.96 | 72.38 | 82.72 | 81.02 | 76.73 | 69.38 | 77.57 | 74.56 |
| DPIC | 75.99 | 72.77 | 83.51 | 77.42 | 87.01 | 83.21 | 95.60 | 88.61 |
| Roberta-base | **89.43** | 72.89 | 81.81 | 81.38 | **90.90** | 88.43 | 96.46 | 91.93 |
| **HLD(Ours)** | 88.89 | **79.40** | **91.54** | **86.61** | 90.42 | **89.31** | **97.65** | **92.46** |

domain shift; however, our method consistently achieves the best or second-best AUROC across datasets, with an average of 86.61%, significantly outperforming the next best baseline RoBERTa-base (81.38%). In the cross-model setting (trained on GPT-3.5 and tested on Claude, PaLM-2, and LLaMA-2), our method again ranks first with an average AUROC of 92.46%. Cross-domain evaluation is particularly challenging due to large discrepancies in vocabulary distributions. Leveraging both syntactic and semantic layer distribution, our approach moves beyond shallow text patterns and captures machine-generated signals that transfer across domains. For cross-model generalization, since different LLMs share common generation mechanisms, their outputs are more similar to each other than to human writing. Our framework explicitly measures whether a text aligns more with the shared statistical distribution of AI generations or with the distribution of human writing, enabling robust capture of AI-specific regularities. The complete results in the Appendix **??** demonstrate that our method maintains strong generalization across all domain and model combinations, with smaller performance drops compared to baselines. Furthermore, to validate the effectiveness of HLD on the latest generation of LLMs, we extended our evaluation to include cutting-edge models such as GPT-5, DeepSeek-R1/V3 (Guo et al., 2025; Liu et al., 2024b), and Claude-3.5. As detailed in Appendix B.1, HLD maintains state-of-the-art overall average performance on these new models.

**Robustness Analysis.** In real-world applications, users may employ various strategies to evade detection. We conduct experiments under three attack scenarios to evaluate the robustness of our method, with results reported in Table 3. For baselines that require training, models are trained on the Direct dataset before being evaluated on the attack test sets. Our method again ranks first in average performance with an AUROC of 97.85%, outperforming the second-best method (DPIC) by an absolute margin of 4.82 percentage points. Crucially, under the most challenging paraphrase attacks, where the performance of all baselines drops significantly, HLD maintains a high AUROC of 97.04%, demonstrating HLD's stability and reliability in adversarial environments. This outstanding robustness stems from the inherent advantages of our hierarchical distribution design, which proves highly robust against shallow modifications by modeling deeper linguistic structures that remain stable even when surface-level words are altered. Moreover, by integrating evidence from every single token across the entire text, our method's final judgment is inherently resilient to localized modifications.

**Ablation Study.** To investigate the contribution of each feature component (Word, POS, Dep, Semantic), we conducted an ablation study, training on the Arxiv dataset and testing across all domains, with results shown in Table 4. We can notice that removing any single feature dimension leads to a performance drop. Notably, removing word features (-Word) causes the most significant harm to out-of-domain generalization, with the AUROC on the "Review" domain dropping from 91.45% to 84.43%. Similarly, removing syntactic (-POS, -Dependency) or semantic (-Semantic) features also results in a general performance decline. This clearly demonstrates that our designed word, syntactic, and semantic feature dimensions are all effective and complementary.

**Interpretability.** We conducted a qualitative case study by visualizing the log-likelihood ratio (LLR) of each token as a heatmap. As shown in Figure 4 for an abstract from the Arxiv test set, a

Table 3: Adversarial robustness of all detectors. We report performance on original data (Direct) and against three attack scenarios. Best and second-best results in each column are marked in **bold** and underlined. * denote the zero-shot methods.

| Detector ↓ | Direct | | Paraphrase | | Perturbation | | Data Mixing | | Avg. | |
|---|---|---|---|---|---|---|---|---|---|---|
| | AUROC | F1 | AUROC | F1 | AUROC | F1 | AUROC | F1 | AUROC | F1 |
| LRR* | 85.83 | 77.40 | 63.99 | 55.20 | 45.91 | 29.27 | 66.12 | 53.81 | 65.46 | 54.92 |
| DetectGPT* | 52.84 | 40.90 | 31.79 | 16.89 | 18.21 | 00.00 | 26.28 | 00.00 | 32.28 | 14.45 |
| Binoculars* | 94.87 | 89.73 | 88.34 | 81.56 | 76.89 | 69.34 | 89.12 | 83.67 | 87.31 | 81.08 |
| Fast-DetectGPT* | 79.56 | 72.45 | 70.12 | 62.89 | 49.56 | 41.23 | 67.23 | 59.78 | 66.62 | 59.09 |
| DNA-GPT* | 88.01 | 80.78 | 65.61 | 54.94 | 40.45 | 02.73 | 62.14 | 50.89 | 64.05 | 47.34 |
| RADAR | 92.19 | 76.01 | 57.87 | 59.98 | 90.24 | 76.26 | 96.31 | 80.31 | 84.15 | 73.14 |
| Ghostbuster | 88.48 | 80.22 | 62.70 | 66.69 | 87.05 | 79.30 | 88.77 | 80.69 | 81.75 | 76.73 |
| RAIDAR | 91.15 | 82.72 | 66.40 | 59.97 | 91.40 | 83.37 | 88.35 | 80.08 | 84.33 | 76.54 |
| DPIC | 98.02 | 89.69 | 76.84 | 55.25 | **99.65** | **96.82** | 97.62 | 88.04 | 93.03 | 82.45 |
| Roberta-base | 95.68 | 90.06 | 73.00 | 73.35 | 98.40 | 94.23 | 93.82 | 87.19 | 90.23 | 86.21 |
| **HLD(Ours)** | **98.91** | **96.39** | **97.04** | **93.05** | 98.77 | 95.80 | 96.69 | **91.89** | **97.85** | **94.28** |

Table 4: Ablation study on the contribution of each component. The model is trained on the Arxiv dataset and evaluated across all four domains.

| Configuration | Arxiv | | XSum | | Writing | | Review | |
|---|---|---|---|---|---|---|---|---|
| | AUROC | F1 | AUROC | F1 | AUROC | F1 | AUROC | F1 |
| HLD (all) | **99.45** | **97.31** | **88.89** | **81.47** | **79.40** | 71.90 | **91.54** | **83.90** |
| - Word | 98.89 | 95.80 | 87.57 | 79.61 | 77.80 | 69.34 | 84.43 | 79.68 |
| - POS | 99.15 | 96.63 | 85.45 | 78.91 | 75.21 | **72.99** | 88.10 | 81.41 |
| - Dep | 99.19 | 96.89 | 87.16 | 80.87 | 75.58 | 68.30 | 90.05 | 81.49 |
| - Semantic | 99.10 | 96.41 | 88.29 | 80.98 | 76.24 | 68.11 | 87.31 | 80.26 |

yellow background indicates the model deems a token more AI-like, while blue suggests it is more human-like. The analysis reveals that our method identifies "formulaic" academic phrases such as "valuable insights," "findings contribute to," and "demonstrate the reliability" as AI-like. This case study provides visual evidence that our model's decision-making process is transparent and relies on discernible linguistic patterns rather than running as an unexplainable "black box."

**Impact of N-gram Context Length.** Figure 5a presents our analysis of the n-gram context length $k$, revealing that near-optimal performance is attained with remarkably short context lengths, specifically for $k$ between 3 and 5. The figure clearly contrasts the robustness of our hierarchical features. While the performance at the word-level deteriorates sharply as k increases, underscoring its vulnerability to data sparsity, the syntactic and semantic level exhibit exceptional stability. Notably, the syntactic feature sustains a near-perfect AUROC across a wide range of k (from 2 to 10). This result strongly validates our hierarchical design, demonstrating that abstract linguistic signals are not only more robust but also significantly more efficient for discriminating between human-written and machine-generated text than volatile shallow-level patterns.

**Impact of Text Length.** The length of a text critically affects detection performance, with longer texts yielding richer reliable signals to differentiate human-written and machine-generated content. Figure 5b shows that all methods benefit from increased text length. Remarkably, our method HLD significantly outperforms both the supervised (RoBERTa-base) and zero-shot (Fast-DetectGPT) baselines on short texts (<50 words), achieving over 80% AUC, which is critical for real-world application like social media analysis. Moreover, the performance of HLD saturates at a high level (around 99% AUROC) once the text length exceeds approximately 150 words. These results benefits from hierarchical token-by-token distribution analysis, making HLD exhibit strong adaptability and superior performance across texts of varying lengths.

**Impact of Data Scale.** The relationship between model performance and the library size used to construct the databases are presented in Figure 5c. The experiment indicates that model performance improves rapidly with increasing data, achieving over 95% AUROC with only 20k words

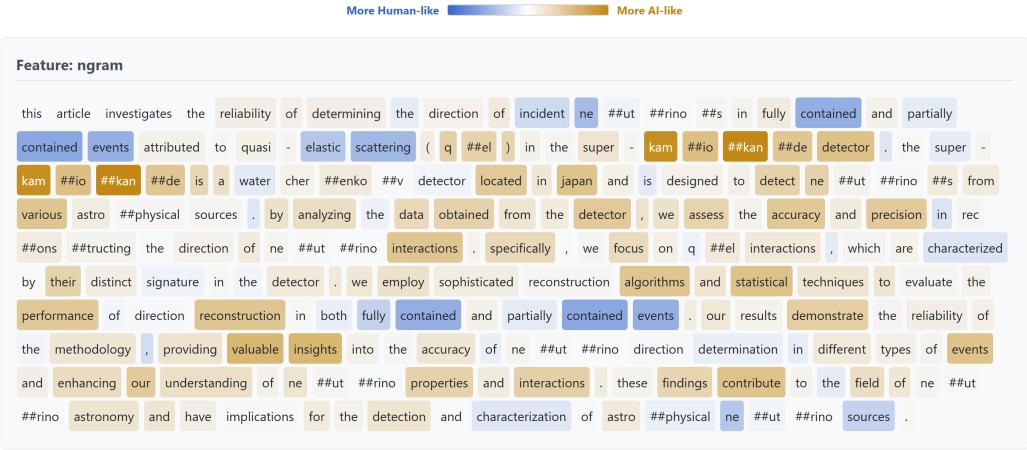

Figure 4: Word-level attribution visualization for an AI-generated abstract from the Arxiv test set. Tokens are colored by their log-likelihood ratio (LLR): yellow for AI-like and blue for human-like. See Appendix B.1 for analyses at other feature granularities.

and plateauing at roughly 100k words. This demonstrates that our model is highly data-efficient and attains strong performance without massive data, which is an important advantage for rapid deployment in specific domains.

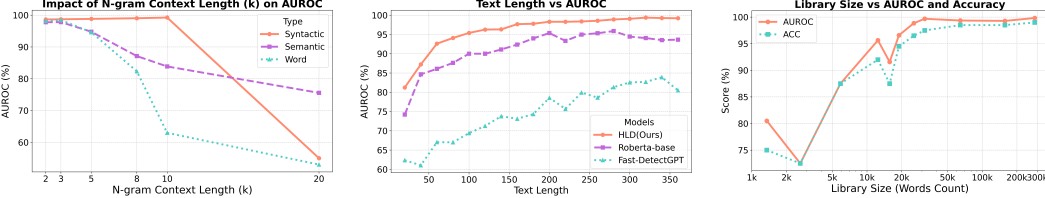

(a) Impact of N-gram Context Length $k$ on AUROC.

(b) Impact of Text Length on AU-ROC.

(c) Impact of Library Size on Performance.

Figure 5: Analysis of the impact of key hyperparameters and data conditions on detector performance. (a) Discriminative signals are effectively captured with $k \leq 5$. (b) HLD consistently outperforms baselines across all text lengths. (c) HLD demonstrates high data efficiency, achieving strong performance with a relatively small library size.

**Evaluation on Non-English Dataset.** To demonstrate effectiveness of HLD beyond English, we test our approach and the baselines on the Chinese and Russian datasets from the M4 benchmark (Wang et al., 2024), and the Korean dataset from KatFishNet (Park et al., 2025).

As shown in Figure 6, HLD attains the highest detection performance across all three languages, attaining AUROC scores of 98.41%, 95.84%, and 84.97%, respectively. Notably, HLD exhibits marked superiority over the two baseline methods (RoBERTa and Fast-DetectGPT). When compared to RoBERTa, HLD outperforms it by margins of 4.32% on Chinese, 10.55% on Russian, and 8.99% on Korean, validating that the statistical features captured by our approach are effective across diverse linguistic contexts.

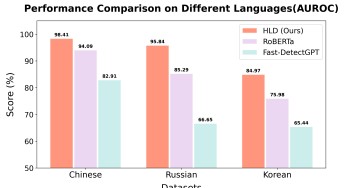

Figure 6: Performance of our method on the Chinese dataset.

**Efficiency Analysis.** Beyond detection accuracy, efficiency is important for practical use. We evaluate inference latency and peak GPU memory on 100 test samples using a single NVIDIA A800 (80GB). As summarized in Table 5, Our framework achieves an inference time of 5.24 seconds and a memory footprint of 2644 MiB, while Fast-DetectGPT and RAIDAR incur significantly higher costs, as they rely on proxy models for inference. Importantly,

HLD preserves the SOTA performance reported in earlier sections, but requiring a lower computational budget, making it more practical for real-world deployment.

Table 5: Efficiency comparison of different detection methods. We report the average inference time (per document) and peak GPU memory usage on the test set.

| Metric | Ours | RoBERTa-base | Fast-DetectGPT | RAIDAR |
|---|---|---|---|---|
| Total Time (s) | 5.24 | 2.14 | 35.13 | 3264.8 |
| GPU Memory Usage (MiB) | 2644 | 1056 | 36154 | 23594 |

## 4 CONCLUSION

We presented HLD, a hierarchical linguistic distribution framework for detecting LLM-generated text. By modeling word, syntactic, and semantic distributions with log likelihood ratios, HLD achieves strong performance across domains, models, and adversarial settings, while offering interpretability and efficiency advantages over large-model-based detectors. The method shows particular strength on short texts, requires modest data and computational resources, and provides intuitive visual explanations.

## EHICS STATEMENT

This work explores hierarchical linguistic distribution modeling for detecting LLM-generated text. While effective in our experiments, the method is not foolproof and may be circumvented by paraphrasing or more advanced models. Its outputs should be regarded as probabilistic signals rather than conclusive proof, and never as the sole basis for high-stakes decisions; careful human oversight remains essential. We further caution against misuse for surveillance or censorship, and recommend responsible deployment that respects privacy and academic freedom.

## REPRODUCIBILITY STATEMENT

To ensure the reproducibility of our study, we provide implementation details and specific parameter settings of our method in Section C.1. Moreover, all datasets used in this research are publicly available and have been employed in prior work in the field.

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

# A

## A.1 RELATED WORK

**Machine Text Generation.** Modern large language models (LLMs) are predominantly autoregressive Transformers trained with next-token prediction on large text corpora (Vaswani et al., 2017; Radford et al., 2019; Brown et al., 2020). Given a tokenized sequence $x_{1:T}$, the model parameterized by $\theta$ learns the conditional distribution $p_\theta(x_t \mid x_{<t})$ by minimizing the cross-entropy (negative log-likelihood)

$$\mathcal{L}_{\text{LM}}(\theta) = -\sum_{t=1}^{T} \log p_\theta(x_t \mid x_{<t}).$$

Scaling data, parameters, and compute follows well-documented scaling laws, which guide compute-optimal training and early-stopping strategies (Kaplan et al., 2020; Hoffmann et al., 2022). Decoder-only models (e.g., GPT-2/3)(Radford et al., 2019; Brown et al., 2020) and their open counterparts (e.g., LLaMA, Deepseek)(Touvron et al., 2023; Liu et al., 2024b) have therefore become the dominant architecture for open-ended text generation (Achiam et al., 2023). At generation time, decoding strategy shapes the output distribution and style: while beam search can induce repetition and degeneration, stochastic methods such as top-$k$ (Fan et al., 2018) and nucleus (top-$p$) sampling (Holtzman et al., 2020) improve diversity and fluency. Beyond maximum-likelihood pretraining, instruction tuning and reinforcement learning from human feedback (RLHF) further align the model's conditional distribution with human preferences, improving helpfulness and safety while altering likelihood patterns later exploited by detectors (Wei et al., 2021; Ouyang et al., 2022).

**Detect Machine-Generated Text.** We categorize prior work on LLM-generated text detection into watermarking, supervised classifiers, and zero/low-shot statistical methods.

Model-time watermarking perturbs decoding to embed a verifiable statistical signal. The green-list scheme of (Kirchenbauer et al., 2023) biases a keyed token subset and enables a simple hypothesis test without model access, establishing a widely used baseline. (Hou et al., 2024) proposed a robust sentence-level semantic watermarking algorithm and (Liu et al., 2024a) proposed unforgeable publicly verifiable watermark. However, recent work (Cheng et al., 2025) achieves successful attacks across various watermarking algorithms. Beyond specific attacks, watermarks also face challenges such as reduced text quality and barriers to practical deployment.

Supervised classifiers formulate detection as a binary classification problem, training models on extensive datasets of both human and machine-generated text. A common strategy is to fine-tune pre-trained language models such as RoBERTa(Liu et al., 2019)) with an added classification head. Some methods focus on engineering linguistic features (e.g., sentence length, repetition patterns)(Gallé et al., 2021; Yadagiri et al., 2024). Additionally, Ghostbuster (Verma et al., 2024) extracts features from models and trains a classifier to identify AI-generated content. DPIC(Yu et al., 2024) and RAIDAR(Mao et al., 2024) use large models to regenerate the text and analyze features for detection. As a result, these detectors typically achieve strong performance when the source of machine-generated text is represented in training data. However, a critical and widely acknowledged limitation of supervised methods is their lack of generalizability (Li et al., 2024) and interpretability.

Zero-shot detectors avoid task-specific training. GLTR(Gehrmann et al., 2019) visualizes token-rank statistics to aid detection. DetectGPT(Mitchell et al., 2023) proposes a curvature test over a model's log-probability surface; Fast-DetectGPT(Bao et al., 2023) replaces perturbation sampling with conditional-probability curvature, cutting cost dramatically. Binoculars(Hans et al., 2024) scores passages via paired-model contrast. More recently, (Bao et al., 2025) aims to estimate full distributions based on partial observations from API-based models, and Lastde++ (Xu et al., 2025) identifies MGT by mining token probability sequences. Despite these advances, zero-shot methods continue to face challenges of low accuracy and unstable performance in certain scenarios.

## A.2 DETAILS OF THE BASELINES

**LRR** (Su et al., 2023) leverages log-rank information to distinguish between human and machine-generated text. The core idea is that the distribution of word ranks in a text produced by a large language model (LLM) will have different statistical properties than a text written by a human.

**DetectGPT** (Mitchell et al., 2023) is a zero-shot method based on the hypothesis that LLM-generated text lies in areas of negative curvature in the model's log-probability function. It compares the log probability of original text with perturbations, using significant differences to indicate machine generation. The T5-small model is used as the perturbation model, with the number of perturbations set to 10.

**Fast-DetectGPT** (Bao et al., 2023) optimizes DetectGPT by replacing multiple perturbations with a more efficient sampling strategy, improving speed and detection accuracy. We directly adopt the settings from DetectRL, using GPT-Neo-2.7B as the scoring model and GPT-J-6B as reference model.

**Binoculars** (Hans et al., 2024) is a zero-shot detector with a low false positive rate, using two pre-trained models: an observer and a performer. It compares the perplexity of the text evaluated by the observer and the cross-perplexity of the performer's predictions to identify AI-generated text. GPT-Neo-2.7B is used for computing perplexity, while GPT-J-6B is employed for calculating cross-perplexity.

**DNA-GPT** (Divergent N-Gram Analysis) (Yang et al., 2024) truncates text and uses an LLM to regenerate the remainder. It analyzes the divergence between the original and generated text using n-gram analysis, revealing differences between human and machine-generated text. We use GPT-4o-mini as a continuation tool. The default settings are a truncation rate of $\tau = 0.5$ and 10 rewrites.

**Lastde++** (Xu et al., 2025) treats Token Probability Sequences (TPS) as time series, analyzing local dynamics. It finds that human text exhibits abrupt fluctuations in token probabilities, while LLM-generated text is smoother and more predictable. By default, we use GPT-J-6B as the scoring model.

**RADAR** (Robust AI-text Detection via Adversarial leaRning) (Hu et al., 2023) trains a paraphraser and a detector in an adversarial setup, making the detector robust to paraphrasing attacks and more effective in identifying AI-generated text. We directly use the official RADAR-Vicuna-7B model.

**Ghostbuster** (Verma et al., 2024) detects text from black-box LLMs by passing the document through weaker, publicly available models. It extracts features and trains a classifier to identify AI-generated content, making it practical for real-world scenarios. Following the original work, we extract token probabilities from four models: a unigram and a trigram model trained on the Brown Corpus, alongside GPT-3 ada and davinci. Features are generated by a structured search combining these probability vectors with predefined operations. A Logistic Regression classifier is then trained on the selected features to make the final prediction.

**RAIDAR** (geneRative AI Detection viA Rewriting) (Mao et al., 2024) detects AI text by comparing its edit distance after being rewritten by an LLM. AI text shows a smaller edit distance compared to human text. The details of our implementation are shown in the Table 6.

Table 6: Implementation details for the Raidar baseline (Mao et al., 2024).

| Component | Specification |
| --- | --- |
| *Feature Generation* | |
| Rewriting LLM | GPT-3.5-Turbo |
| Distance Metric | Normalized Levenshtein Score |
| Invariance Prompts | 1. 'Help me polish this:' |
| | 2. 'Rewrite this for me:' |
| | 3. 'Refine this for me please:' |
| Equivariance T-Prompts | 1. 'Write this in the opposite meaning:' |
| | 2. 'Rewrite to Expand this:' |
| Equivariance $T^{-1}$-Prompts | 1. 'Write this in the opposite meaning:' |
| | 2. 'Rewrite to Concise this:' |
| Classifier | Logistic Regression or XGBoost |

**DPIC** (Decoupling Prompt and Intrinsic Characteristics) (Yu et al., 2024) detects black-box model text by reconstructing the prompt and comparing the original text with new text generated from it.

This isolates the generative model's "fingerprint". The full replication details are provided in Table 7.

Table 7: Replication details for the DPIC baseline, following the original work (Yu et al., 2024).

| Parameter | Specification |
|---|---|
| *Model Architecture* | |
| Encoder | RoBERTa (weights frozen)(Liu et al., 2019) |
| Classifier | 3-layer MLP |
| *Training Strategy* | |
| Optimizer | Adam |
| Learning Rate | $1 \times 10^{-3}$ |
| Epochs | 10 |
| Loss Function | Binary Cross-Entropy (BCE) |
| Model Selection | Highest AUROC on validation set |

**Roberta-base** (Liu et al., 2019) serves as a baseline for supervised AI text detectors. Fine-tuned on human and AI-generated text, it provides a strong benchmark for evaluating detection methods. The training parameters are as follows: the learning rate is set to $1e^{-6}$ with the AdamW optimizer and a weight decay of $1e^{-4}$; the number of training epochs is 5, with an early stopping strategy that halts training when the F1 score on the validation set decreases.

## B

### B.1 SUPPLEMENTARY TO MAIN RESULTS

Table 8: Performance comparison of all evaluated detectors across four large language models and four distinct text domains. We report F1 scores (in %). The best and second-best scores in each column are highlighted in **bold** and underlined, respectively. *denote the zero-shot methods.

| Detector | Large Language Models | | | | | Text Domains | | | | |
|---|---|---|---|---|---|---|---|---|---|---|
| | GPT-3.5 | Claude | PaLM-2 | Llama-2 | Avg. | Arxiv | XSum | Writing | Review | Avg. |
| LRR* | 52.12 | 18.91 | 65.51 | 75.51 | 53.01 | 61.34 | 38.38 | 53.09 | 68.99 | 55.45 |
| DetectGPT* | 26.27 | 12.56 | 00.00 | 20.40 | 14.81 | 00.00 | 00.00 | 50.83 | 35.25 | 21.52 |
| Binoculars* | 82.50 | 39.35 | 88.20 | 92.30 | 75.59 | 76.77 | 72.18 | 79.73 | 84.32 | 78.25 |
| Fast-DetectGPT* | 59.55 | 00.00 | 57.58 | 69.08 | 46.55 | 24.46 | 28.39 | 67.84 | 71.62 | 48.08 |
| DNA-GPT* | 55.04 | 25.67 | 60.77 | 62.89 | 51.09 | 22.07 | 65.30 | 65.05 | 62.06 | 53.62 |
| Lastde++* | 77.07 | 67.45 | 77.72 | 81.33 | 75.89 | 75.84 | 72.35 | 75.94 | 77.87 | 75.50 |
| RADAR | 68.94 | 49.42 | 82.67 | 81.26 | 70.57 | 61.59 | 91.68 | 54.19 | 75.28 | 70.69 |
| Ghostbuster | 83.70 | 75.37 | 73.59 | 81.26 | 78.48 | 84.14 | 86.60 | 76.70 | 78.10 | 81.39 |
| RAIDAR | 82.36 | 80.36 | 82.69 | 81.45 | 81.72 | 87.98 | 88.89 | 80.37 | 88.06 | 86.33 |
| DPIC | 96.61 | 86.27 | 87.58 | 87.81 | 89.57 | 94.31 | 95.39 | 85.52 | 91.79 | 91.75 |
| Roberta-base | 97.11 | 93.54 | 90.34 | 94.08 | 93.77 | **100.00** | **98.80** | 90.78 | 95.93 | 96.38 |
| **HLD(Ours)** | **98.40** | **97.01** | **93.06** | **97.09** | **96.39** | 97.31 | 97.29 | **98.11** | **98.70** | **97.85** |

**Overall Performance (f1 score).** Table 8 reports the F1 scores (%) of twelve detectors evaluated on four LLMs (GPT-3.5, Claude, PaLM-2, Llama-2) and four text domains (ArXiv, XSum, Writing, Review). Our proposed method, HLD, achieves the highest mean F1 both across models (96.39%) and across domains (97.85%). Relative to the strongest non-HLD baseline (RoBERTa-base), HLD improves the averages by +2.62 points across models (96.39 vs. 93.77) and +1.47 points across domains (97.85 vs. 96.38). Importantly, HLD is the only detector that consistently exceeds 93% F1 across all four models (98.40/97.01/93.06/97.09), demonstrating stable performance under varying model sources. On the domain side, while RoBERTa-base achieves strong results on ArXiv and XSum, it degrades substantially on Writing (90.78%). In contrast, HLD maintains balanced performance across domains, obtaining the best scores on Writing (98.11%) and Review (98.70%) while remaining competitive on ArXiv and XSum. Overall, HLD attains the strongest average per-

formance with reduced variance, highlighting its robustness across heterogeneous models and text domains.

Table 9: Cross-domain generalization performance of all detectors, measured by AUROC (in %). Each block shows results for a model trained on a specific source domain and evaluated on the others. For readability, the table is in two parts. The best and second-best scores are highlighted in **bold** and underlined, respectively. *denote the zero-shot methods.

| Detector↓ Eval→ | Train on Arxiv | | | | Train on XSum | | | |
|---|---|---|---|---|---|---|---|---|
| | XSum | Writing | Review | **Avg.** | Arxiv | Writing | Review | **Avg.** |
| LRR* | 40.88 | 38.44 | 55.81 | 45.04 | 57.45 | 39.08 | 55.81 | 50.78 |
| Fast-DetectGPT* | 23.71 | 59.67 | 60.17 | 47.85 | 28.43 | 62.99 | 63.08 | 51.50 |
| GhostBuster | 85.81 | 75.18 | 69.70 | 76.90 | 86.32 | 63.27 | 68.57 | 72.72 |
| RAIDAR | 87.96 | 72.38 | 82.72 | 81.02 | 85.06 | 60.24 | 64.73 | 70.01 |
| DPIC | 75.99 | 72.77 | 83.51 | 77.42 | 89.36 | **87.03** | 88.54 | 88.31 |
| Roberta-base | **89.43** | 72.89 | 81.81 | 81.38 | 89.56 | 78.08 | 76.28 | 81.31 |
| **HLD(Ours)** | 88.89 | **79.40** | **91.54** | **86.61** | **90.34** | 86.86 | **89.92** | **89.04** |

| Detector↓ Eval→ | Train on Writing | | | | Train on Review | | | |
|---|---|---|---|---|---|---|---|---|
| | Arxiv | XSum | Review | **Avg.** | Arxiv | XSum | Writing | **Avg.** |
| LRR* | 61.14 | 46.31 | 67.98 | 58.48 | 61.49 | 47.02 | 57.12 | 55.21 |
| Fast-DetectGPT* | 34.81 | 33.06 | 68.30 | 45.39 | 40.70 | 37.66 | 68.25 | 48.87 |
| GhostBuster | 85.12 | 87.26 | 75.71 | 82.70 | 79.23 | 85.23 | 84.47 | 82.98 |
| RAIDAR | 77.09 | 62.08 | 86.73 | 75.30 | 77.41 | 69.90 | 78.81 | 75.37 |
| DPIC | 87.87 | 83.11 | 94.55 | 88.51 | 88.44 | 80.60 | 94.04 | 87.69 |
| Roberta-base | 86.69 | 78.95 | 95.41 | 87.02 | 77.60 | 79.03 | 95.50 | 84.04 |
| **HLD(Ours)** | **87.94** | **93.82** | **95.79** | **92.52** | **89.42** | **93.65** | **96.42** | **93.16** |

**Complete generalization results.** Tables 9–10 report the full AUROC (%) results under cross-domain and cross-model transfer settings. In the cross-domain case, detectors are trained on one source domain (ArXiv/XSum/Writing/Review) and evaluated on the remaining domains, while in the cross-model case, training is performed on one source LLM (GPT-3.5/Claude/PaLM-2/Llama-2) and evaluation on the others. The results show that although all methods are affected by distribution shifts, HLD consistently achieves the highest averages across all source domains, for instance reaching 86.61% when trained on ArXiv compared to 81.31% for RoBERTa-base, and delivering particularly strong gains when trained on Writing and Review, with averages of 92.52% and 93.16%. Similarly, in the cross-model setting HLD again outperforms all baselines for every source LLM, achieving 92.46% (GPT-3.5), 95.14% (Claude), 97.28% (PaLM-2), and 96.72% (Llama-2). Overall, the complete tables corroborate the main-text findings: HLD maintains consistently strong detection ability across diverse source–target combinations and exhibits smaller performance drops than the strongest baseline, underscoring its robustness in both cross-domain and cross-model transfer scenarios.

**Evaluations on Latest Models.** To verify the effectiveness of HLD on the latest generation of Large Language Models, we followed the DetectRL(Wu et al., 2024) setting and extended our evaluation to include cutting-edge models such as GPT-5, DeepSeek-R1/V3, and Claude-3.5. As shown in Table 11, when trained and tested on the same new model, HLD demonstrates exceptional adaptability. It consistently achieves near-perfect detection scores (e.g., 99.63% AUROC on DeepSeek-R1 and 99.54% on GPT-5), significantly outperforming zero-shot baselines and surpassing the strong supervised method RoBERTa and DPIC. This indicates that HLD can effectively capture the generative signatures of even the most modern LLMs. A more challenging and practical scenario is detecting text from new models using detectors trained only on older models. We simulated this by training on data generated by GPT-3.5 and testing on data from newer targets. HLD maintains high detection capability across all new models, achieving the highest average AUROC of 92.84%. Notably, on GPT-5, HLD retains an F1 score of 88.03%, outperforming RoBERTa by nearly 8 points. This con-

Table 10: Cross-model generalization performance of all detectors, measured by AUROC (in %). Each block shows results for a model trained on a specific source model and evaluated on the others. For readability, the table is in two parts. The best and second-best scores are highlighted in **bold** and underlined, respectively. * denote the zero-shot methods.

| Detector↓ Eval→ | Train on GPT-3.5 | | | | Train on Claude | | | |
|---|---|---|---|---|---|---|---|---|
| | Claude | PaLM-2 | Llama-2 | **Avg.** | GPT-3.5 | PaLM-2 | Llama-2 | **Avg.** |
| LRR* | 24.70 | 61.79 | 75.34 | 53.94 | 45.73 | 57.66 | 72.67 | 58.69 |
| Fast-DetectGPT* | 12.96 | 59.56 | 69.93 | 47.48 | 00.19 | 00.00 | 01.18 | 00.46 |
| GhostBuster | 61.83 | 77.43 | 83.66 | 74.31 | 71.39 | 79.99 | 78.00 | 76.46 |
| RAIDAR | 76.73 | 69.38 | 77.57 | 74.56 | 78.75 | 76.71 | 76.80 | 77.42 |
| DPIC | 87.01 | 83.21 | 95.60 | 88.61 | 98.80 | 84.51 | 93.84 | 92.38 |
| Roberta-base | **90.90** | 88.43 | 96.46 | 91.93 | 98.33 | **91.63** | 95.20 | 95.05 |
| **HLD(Ours)** | 90.42 | **89.31** | **97.65** | **92.46** | **98.86** | 90.80 | **95.77** | **95.14** |

| Detector↓ Eval→ | Train on PaLM-2 | | | | Train on Llama-2 | | | |
|---|---|---|---|---|---|---|---|---|
| | GPT-3.5 | Claude | Llama-2 | **Avg.** | GPT-3.5 | Claude | PaLM-2 | **Avg.** |
| LRR* | 52.36 | 26.23 | 75.58 | 51.39 | 52.14 | 25.25 | 62.23 | 46.54 |
| Fast-DetectGPT* | 55.77 | 08.20 | 68.43 | 44.13 | 56.28 | 08.65 | 57.74 | 40.89 |
| GhostBuster | 79.48 | 79.49 | 85.30 | 81.42 | 84.51 | 76.78 | 81.73 | 81.01 |
| RAIDAR | 75.26 | 76.19 | 83.49 | 78.31 | 86.86 | 77.46 | 86.43 | 83.58 |
| DPIC | 99.28 | 89.17 | 98.54 | 95.66 | 99.24 | 86.61 | 90.33 | 92.06 |
| Roberta-base | 98.76 | 93.39 | 98.26 | 96.80 | 98.39 | 91.69 | 93.06 | 94.38 |
| **HLD(Ours)** | **99.61** | **93.64** | **98.60** | **97.28** | **99.71** | **94.78** | **95.66** | **96.72** |

Table 11: Comprehensive performance comparison of the latest models. The top section presents methods evaluated under in-model (in-domain) or zero-shot settings. The bottom section presents detectors trained on GPT-3.5 (cross-model) evaluated on the target LLMs. Best and second-best scores in each column within each section are highlighted in **bold** and underlined, respectively. * denotes zero-shot methods.

| Detector ↓ | DeepSeek-R1 | | GPT-5 | | Claude-3.5 | | Qwen3-Max | | GPT-4 | | DeepSeek-V3 | | Avg. | |
|---|---|---|---|---|---|---|---|---|---|---|---|---|---|---|
| | AUROC | F1 | AUROC | F1 | AUROC | F1 | AUROC | F1 | AUROC | F1 | AUROC | F1 | AUROC | F1 |
| **In-Model / Zero-Shot Settings** | | | | | | | | | | | | | | |
| Fast-DetectGPT* | 59.28 | 49.43 | 63.32 | 52.55 | 61.65 | 52.59 | 75.39 | 64.23 | 63.74 | 51.24 | 66.75 | 59.26 | 65.02 | 54.88 |
| Binoculars* | 82.13 | 59.50 | 87.63 | 69.92 | 89.10 | 76.04 | 96.75 | 91.14 | 92.78 | 81.40 | 92.09 | 83.29 | 90.08 | 76.88 |
| Lastde++* | 65.57 | 68.70 | 83.46 | 74.02 | 88.40 | 77.94 | 94.18 | 86.47 | 88.55 | 77.38 | 86.88 | 77.62 | 84.51 | 77.02 |
| RAIDAR | 93.79 | 87.21 | 88.96 | 81.01 | 90.64 | 84.38 | 90.03 | 81.64 | 91.25 | 82.89 | 92.54 | 85.11 | 91.20 | 83.71 |
| DPIC | 99.22 | 95.90 | 98.78 | 95.31 | 97.03 | 92.41 | 98.30 | 92.98 | **99.54** | 95.67 | 99.52 | 94.73 | 98.73 | 94.50 |
| RoBERTa-base | 99.52 | 96.52 | **99.61** | **97.63** | 98.97 | **98.04** | 99.06 | 95.49 | 98.17 | 94.01 | 98.98 | 94.71 | 99.05 | **96.07** |
| **HLD (Ours)** | **99.63** | **97.87** | 99.54 | 97.41 | **99.48** | 97.43 | **99.43** | **97.29** | 99.14 | **97.10** | **99.67** | **98.08** | **99.48** | 97.53 |
| **Cross-model (Trained on GPT-3.5)** | | | | | | | | | | | | | | |
| RAIDAR | 86.06 | 80.56 | 86.61 | 78.90 | 91.44 | 85.64 | 87.26 | 81.16 | 90.87 | 84.69 | 90.35 | 85.59 | 88.77 | 82.76 |
| DPIC | 83.65 | 80.82 | 90.70 | 83.95 | 96.85 | 92.22 | 92.95 | 87.79 | 96.64 | 92.64 | **90.97** | **86.98** | 91.96 | 87.40 |
| RoBERTa-base | 77.04 | 68.22 | 88.64 | 80.38 | 96.89 | 92.47 | 91.38 | 86.23 | 96.42 | 90.68 | 88.60 | 84.37 | 89.83 | 83.73 |
| **HLD (Ours)** | **87.46** | **84.51** | **92.35** | **88.03** | **97.07** | **95.70** | **94.25** | **90.41** | **96.88** | 92.70 | 89.04 | 80.33 | **92.84** | **88.61** |

firms that HLD learns intrinsic invariants of AI-generated text that persist across model generations, rather than overfitting to the specific artifacts of older models like GPT-3.5.

**Additional visualizations.** In the main text we presented one representative visualization of token-level log-likelihood ratios (LLR). For completeness, Figures 7a 7b 7c provide additional examples using different hierarchy views, including POS n-gram, dependency n-gram, and semantic representations. The color scheme follows the same convention: yellow indicates tokens judged as more AI-like, whereas blue denotes more human-like tokens. These supplementary figures further illustrate that our method consistently highlights formulaic expressions and structural patterns characteristic of machine-generated text, while assigning more human-like scores to contextually varied or domain-specific tokens. Together, these results reinforce the interpretability of our detector and

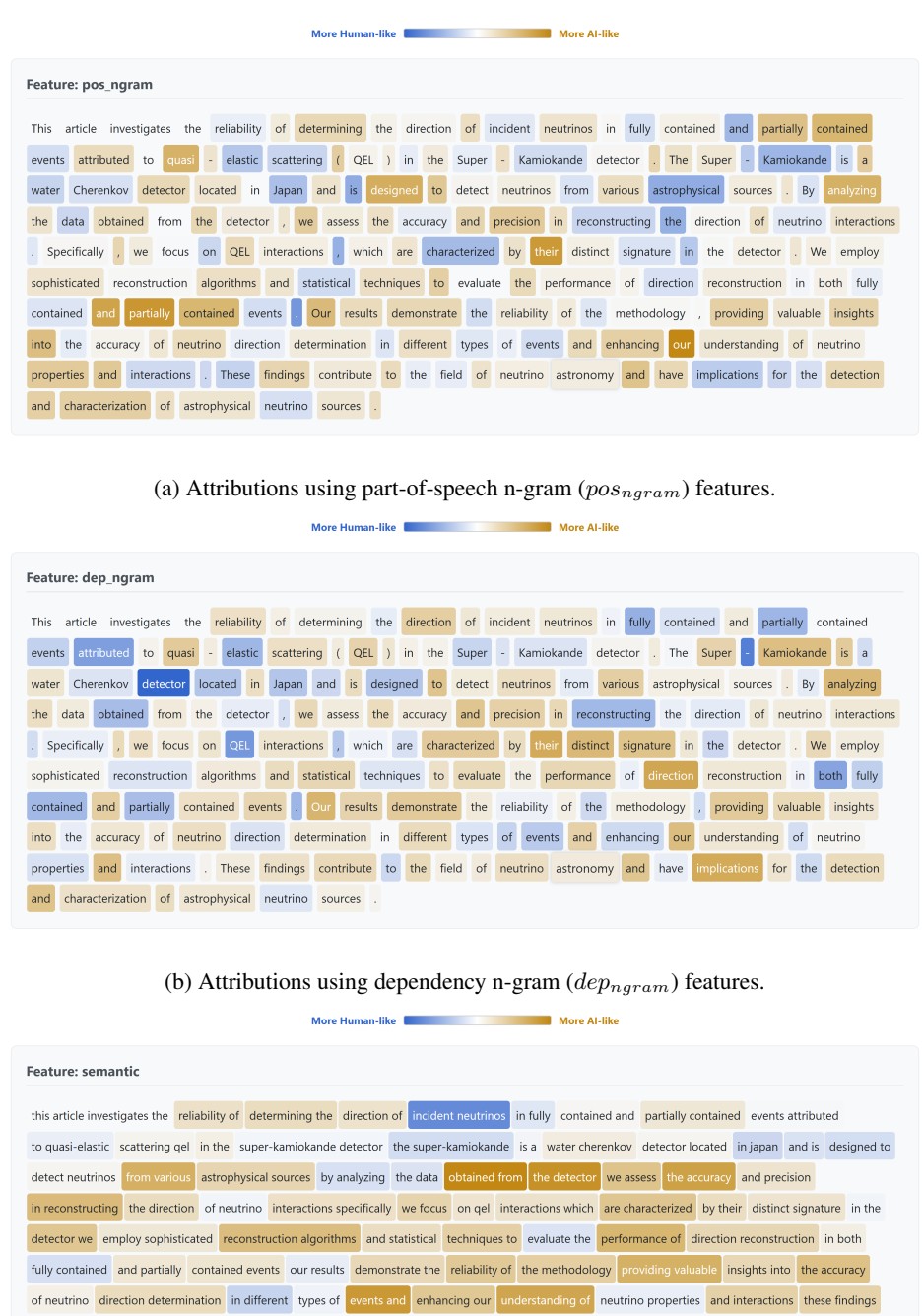

(a) Attributions using part-of-speech n-gram ($pos_{ngram}$) features.

(b) Attributions using dependency n-gram ($dep_{ngram}$) features.

(c) Attributions using semantic features.

Figure 7: Visualization of feature attributions for text detection. The model highlights text portions it identifies as more human-like (blue) or AI-like (orange) based on (a) part-of-speech, (b) dependency, and (c) semantic features.

demonstrate that its predictions are grounded in observable linguistic cues across multiple representational layers.

Table 12: Performance of three-category classification (Human vs. GPT-5 vs. DeepSeek-R1). We report Precision (%) and Recall (%) for each source category. The proposed HLD pipeline utilizes a cascade of binary classifiers (AI vs. Human followed by Source Identification). Best scores are highlighted in **bold**.

| Metric | Detector | Source Category | | |
|--------|----------|-------|-------|------------|
| | | **Human** | **GPT-5** | **DeepSeek-R1** |
| Precision | RoBERTa-base | **99.26** | 79.21 | 77.98 |
| | **HLD (Ours)** | 99.21 | **93.93** | **95.28** |
| Recall | RoBERTa-base | **100.00** | 76.20 | 78.60 |
| | **HLD (Ours)** | **100.00** | **94.40** | **92.90** |

**Extension to Source Attribution.** While our main experiments focus on the binary distinction between machine-generated and human-written text, identifying the specific source model (Model Attribution) is increasingly important for content tracking. To evaluate HLD's capability in this fine-grained task, we implemented a cascaded classification pipeline. Specifically, we constructed a three-category detection framework targeting Human, GPT-5, and DeepSeek-R1. The pipeline consists of two stages: (i) a binary Human vs. AI detector, followed by (ii) a GPT-5 vs. DeepSeek-R1 source identifier for samples predicted as AI. As shown in Table 12, HLD achieves remarkable performance across all three categories. While the baseline RoBERTa-base struggles to distinguish between the two LLMs (dropping to 76-79% in precision and recall for GPT-5 and DeepSeek-R1), HLD maintains high accuracy, achieving over 93% Precision and Recall for both specific AI sources. This result demonstrates that our hierarchical linguistic features capture not only the general discrepancy between human and AI text but also the subtle, model-specific fingerprints inherent to different LLM architectures.

## B.2 RESULTS ON THE OTHER BENCHMARKS.

**Evaluation on the MGTbench.** To provide a comprehensive and impartial evaluation of our proposed HLD (Ours) method, we further conduct extensive experiments on the MGTBench (He et al., 2024) framework. MGTBench is a holistic benchmark designed specifically for assessing the performance of machine-generated text (MGT) detectors against a wide array of powerful Large Language Models (LLMs) and diverse text corpora. Our experimental setup strictly follows the protocol established in the MGTBench paper. We perform our evaluation on its three core datasets: Essay, WP (WritingPrompts), and Reuters. On these datasets, we assess HLD's capability to detect text generated by six distinct LLMs—namely ChatGLM, Dolly, ChatGPT-turbo, GPT4All, StableLM, and Claude. The performance is directly compared against the extensive suite of baseline methods included in MGTBench, such as Log-Likelihood, GLTR, and DetectGPT. The detailed results are presented in Table 13. The data clearly demonstrates that our HLD method achieves remarkably consistent and superior performance across all datasets and LLMs. In the vast majority of test scenarios, the scores for HLD (highlighted in bold) significantly outperform those of all other baseline methods. This underscores the effectiveness, robustness, and strong generalization capability of our approach across various text styles and underlying model architectures.

**Evaluation on the MIRAGE benchmark.** To verify HLD's robustness on more recent datasets, we extended our evaluation to the MIRAGE benchmark (Fu et al., 2025), which encompasses 17 advanced models (e.g., GPT-4o, Claude-3.5) across five domains. Crucially, we evaluated the HLD model trained solely on DetectRL directly on MIRAGE without any fine-tuning. As shown in Table 14, despite the distribution shift, HLD outperforms recent state-of-the-art method DetectAnyLLM(Fu et al., 2025) across both evaluation settings. HLD achieves an AUROC of **96.95%** on the SIG subset and improves the TPR@5% (detection rate at low false positive) by approximately **9%** compared to the previous best result (86.64% vs. 77.70%). This empirically confirms that HLD effectively captures intrinsic linguistic properties that exhibit strong generalization across diverse contexts spanning a range of domains and models.

Table 13: Comparison of HLD (Ours) with baseline methods from the MGTBench evaluation suite. We report F1-scores for each detector. The best and second-best performing methods for each model and dataset are highlighted in **bold** and underlined, respectively.

| Dataset | Method | ChatGLM | Dolly | ChatGPT-turbo | GPT4All | StableLM | Claude |
|---------|--------|---------|-------|---------------|---------|----------|--------|
| Essay | Log-Likelihood | 0.970 | 0.866 | 0.968 | 0.923 | 0.665 | 0.834 |
| | Rank | 0.740 | 0.737 | 0.915 | 0.843 | 0.667 | 0.772 |
| | Log-Rank | 0.983 | 0.865 | 0.966 | 0.923 | 0.692 | 0.814 |
| | Entropy | 0.806 | 0.683 | 0.874 | 0.699 | 0.566 | 0.771 |
| | GLTR | **0.988** | 0.848 | 0.954 | 0.925 | 0.756 | 0.806 |
| | LRR | 0.982 | 0.810 | 0.925 | 0.904 | 0.748 | 0.746 |
| | NPR | 0.956 | 0.865 | 0.218 | 0.927 | 0.740 | 0.238 |
| | DetectGPT | 0.891 | 0.844 | 0.227 | 0.908 | 0.704 | 0.236 |
| | GPTZero | 0.923 | 0.880 | **0.980** | 0.943 | 0.486 | 0.870 |
| | ConDA | 0.668 | 0.069 | 0.000 | 0.260 | 0.663 | 0.664 |
| | OpenAI-D | 0.921 | 0.724 | 0.353 | 0.863 | 0.774 | 0.009 |
| | ChatGPT-D | 0.923 | 0.630 | 0.742 | 0.815 | 0.491 | 0.057 |
| | **HLD(Ours)** | 0.982 | **0.914** | 0.978 | **0.955** | **0.934** | **0.988** |
| WP | Log-Likelihood | 0.980 | 0.794 | 0.841 | 0.934 | 0.786 | 0.773 |
| | Rank | 0.840 | 0.760 | 0.797 | 0.891 | 0.781 | 0.709 |
| | Log-Rank | 0.985 | 0.807 | 0.819 | 0.929 | 0.832 | 0.751 |
| | Entropy | 0.800 | 0.662 | 0.770 | 0.766 | 0.644 | 0.731 |
| | GLTR | 0.983 | 0.766 | 0.800 | 0.935 | 0.861 | 0.733 |
| | LRR | 0.980 | 0.774 | 0.728 | 0.930 | 0.875 | 0.656 |
| | NPR | 0.970 | 0.801 | 0.352 | 0.905 | 0.764 | 0.521 |
| | DetectGPT | 0.812 | 0.719 | 0.608 | 0.808 | 0.695 | 0.517 |
| | GPTZero | 0.980 | 0.732 | 0.980 | **1.000** | 0.148 | 0.818 |
| | ConDA | 0.585 | 0.039 | 0.075 | 0.674 | 0.667 | 0.000 |
| | OpenAI-D | 0.980 | 0.776 | 0.093 | 0.948 | **0.937** | 0.029 |
| | ChatGPT-D | 0.880 | 0.528 | 0.352 | 0.795 | 0.616 | 0.044 |
| | **HLD(Ours)** | **0.997** | **0.847** | **0.988** | 0.974 | 0.892 | **0.958** |
| Reuters | Log-Likelihood | 0.972 | 0.381 | 0.926 | 0.697 | 0.659 | 0.798 |
| | Rank | 0.650 | 0.413 | 0.847 | 0.665 | 0.635 | 0.648 |
| | Log-Rank | 0.990 | 0.373 | 0.944 | 0.735 | 0.701 | 0.785 |
| | Entropy | 0.477 | 0.553 | 0.703 | 0.668 | 0.620 | 0.694 |
| | GLTR | 0.987 | 0.556 | 0.946 | 0.742 | 0.750 | 0.772 |
| | LRR | 0.992 | 0.590 | 0.948 | 0.796 | 0.766 | 0.715 |
| | NPR | 0.950 | 0.790 | 0.284 | 0.843 | 0.751 | 0.560 |
| | DetectGPT | 0.866 | 0.782 | 0.270 | 0.821 | 0.756 | 0.558 |
| | GPTZero | 0.980 | 0.485 | 0.936 | 0.980 | 0.611 | 0.750 |
| | ConDA | 0.664 | 0.137 | 0.000 | 0.667 | 0.000 | 0.667 |
| | OpenAI-D | 0.985 | 0.713 | 0.954 | 0.900 | 0.903 | 0.000 |
| | ChatGPT-D | 0.968 | 0.650 | 0.931 | 0.898 | 0.617 | 0.019 |
| | **HLD(Ours)** | **0.997** | **0.951** | **0.992** | **1.000** | **0.977** | **0.995** |

**Evaluation against Commercial Detectors.** We extended our evaluation to compare HLD with two commercial black-box detectors: GPTZero and ZeroGPT. Experiments were conducted on 2,000 samples randomly drawn from the RAID (Dugan et al., 2024) dataset, covering Open-source and Closed-source models under different decoding strategies. Table 15 reports the Accuracy@FPR=5%. The results indicate that HLD remains highly competitive with commercial tools. While GPTZero shows a slight advantage on Open-source Chat models (98.4% vs. 92.5%), HLD significantly outperforms both commercial detectors on Non-chat models. Specifically, on Open-source Non-chat models with penalty decoding, HLD achieves 46.7% accuracy compared to 4.8% for GPTZero and 0.3% for ZeroGPT. Furthermore, HLD performs best on Closed-source Non-chat models (74.1%), validating the effectiveness of our hierarchical distribution modeling in handling diverse generative behaviors.

Table 14: Performance comparison on MIRAGE-DIG and MIRAGE-SIG datasets. We report AUROC and TPR@5% scores. The best scores in each column are highlighted in **bold**. * denotes zero-shot methods.

| Detector | MIRAGE-DIG | | MIRAGE-SIG | |
|---|---|---|---|---|
| | AUROC | TPR@5% | AUROC | TPR@5% |
| DNA-GPT | 57.33 | 7.76 | 57.59 | 8.13 |
| Fast-DetectGPT* | 77.68 | 43.10 | 77.06 | 42.00 |
| ImBD | 85.97 | 40.65 | 86.12 | 41.83 |
| DetectAnyLLM | 95.25 | 77.70 | 95.26 | 77.22 |
| **HLD (Ours)** | **96.72** | **86.64** | **96.95** | **86.80** |

Table 15: Performance comparison under different settings (Open/Closed source, Chat/Non-chat models, and Penalty strategies). We report detection accuracy. The best score is highlighted in **bold**.

| | Open-source Models | | | | Closed-source Models | |
|---|---|---|---|---|---|---|
| | Chat Models (Llama-c, Mistral-c, MPT-c) | | Non-chat Models (Mistral, MPT, GPT-2) | | Chat Models (ChatGPT, GPT-4, Cohere) | Non-chat Models (Cohere, GPT-3) |
| **Strategy** | | | Sampling | | | |
| **Penalty** | × | ✓ | × | ✓ | × | × |
| GPTZero | **98.4** | 82.5 | 9.4 | 4.8 | **88.5** | 53.4 |
| ZeroGPT | 97.7 | 72.5 | 16.0 | 0.3 | 65.8 | 72.7 |
| **HLD (Ours)** | 92.5 | **88.9** | **33.1** | **46.7** | 83.7 | **74.1** |

## B.3 FURTHER ANALYSIS OF HLD.

**Analysis of n-gram Database.** To quantitatively analyze the statistical differences between AI and human text generation, we first constructed conditional probability distributions for next-token prediction from our $word_{ngram}$ databases, representing both AI and human corpora. We then computed the Kullback-Leibler (KL) Divergence for all shared contexts to measure the dissimilarity between these two sets of distributions (figure 8). The resulting analysis, visualized in the figure, reveals a distinctly bimodal distribution of KL Divergence values. This structure is characterized by a peak near zero, indicating a large number of contexts where AI and human predictions show high agreement, and a second, prominent peak at a high KL value (20 nats), which signifies frequent and drastic disagreements between the two models. We conclude that these moments of high divergence are the core statistical phenomenon that our detection model leverages; while AI can often mimic human-like predictions, it is these systematic points of disagreement that provide a robust and unambiguous signal, making reliable detection possible. This finding was consistently observed across all feature databases, confirming its robustness.

**Statistical Significance of Components.** To verify that the performance contributions of our hierarchical features are not due to random variance, we conducted a rigorous statistical test. We repeated the ablation experiments 5 times on the XSum dataset and reported the average results with standard deviations. As shown in Table 16, removing any single feature dimension (Word, POS, Dependency, or Semantic) leads to a consistent decrease in AUROC. Crucially, we computed the $p$-values for the performance differences between the full HLD model and each variant. The results indicate that all $p$-values are substantially lower than the standard significance level of 0.05 (specifically, all are $< 0.005$). This statistically confirms that every linguistic level in our framework makes a significant and necessary contribution to the detector's overall performance.

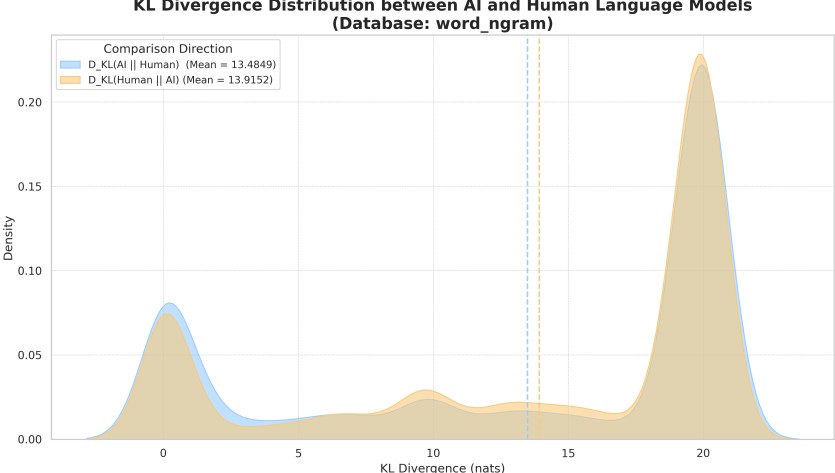

Figure 8: Distribution of KL Divergence between AI and Human language models based on the $word_{ngram}$ database.

Table 16: Statistical significance analysis of ablation studies on the XSum dataset. The experiments were repeated five times to report the mean AUROC ($\pm$ standard deviation). The $p$-values are calculated between the full HLD model and each ablated variant. All differences are statistically significant ($p < 0.01$).

| Configuration | AUROC (%) | $p$-value (vs. Full) |
|---|---|---|
| **HLD (All)** | **$89.07 \pm 0.39$** | – |
| – Word | $87.80 \pm 0.33$ | $4.60 \times 10^{-3}$ |
| – POS | $85.58 \pm 0.25$ | $4.03 \times 10^{-6}$ |
| – Dep | $86.79 \pm 0.35$ | $6.37 \times 10^{-4}$ |
| – Semantic | $88.21 \pm 0.29$ | $3.75 \times 10^{-4}$ |

# C

## C.1 IMPLEMENTATION DETAILS OF HLD

### C.1.1 ALGORITHM OF HLD

We formalize our proposed HLD framework in Algorithm 1. The method first builds hierarchical linguistic distribution databases and then trains a classifier on features derived from the log-likelihood ratios between human and AI text.

### C.1.2 HYPERPARAMETERS AND CONFIGURATION OF HLD

At the word and syntactic levels, we employ n-gram models, setting the context order to $k = 5$. At the semantic level, we estimate the probability of a 2-word target vector given a 4-word context vector encoded by SBERT(all-MiniLM-L6-v2)[4], interpolating from the $M = 4$ nearest neighbors. The neighbor search and indexing are accelerated using faiss-gpu[5]. The specific components and hyperparameters for each level are detailed in Table 17.

The training pipeline uses a 50/50 data split: the first half for distribution estimation (building the statistical databases for MGT and HWT), and the second half for classifier training. We train an XGBoost classifier, optimizing its hyperparameters with Optuna[6] to maximize AUROC over 60

---

[4]See details at https://www.sbert.net.

[5]See details at https://github.com/facebookresearch/faiss.

[6]See details at https://optuna.org.

---

**Algorithm 1** Hierarchical Linguistic Distribution Modeling (HLD)

---

**Require:** Human text corpus $\mathcal{D}_H$, AI text corpus $\mathcal{D}_{AI}$.
**Require:** Set of linguistic levels $\mathcal{J} = \{\text{word, pos, dep, semantic}\}$.
**Require:** Context length parameter (n-gram order) $k$.
**Ensure:** Trained XGBoost classifier $\mathcal{M}_\theta$.
 1: **procedure** BUILDDATABASES($\mathcal{D}_H, \mathcal{D}_{AI}, \mathcal{J}, k$)
 2:     $\mathbb{B} \leftarrow \emptyset$                                                                   ▷ Initialize set of distribution databases
 3:     **for** each linguistic level $j \in \mathcal{J}$ **do**
 4:         $\mathbb{B}_{j,H} \leftarrow$ EstimateDistribution($\mathcal{D}_H, j, k$)     ▷ e.g., N-gram tables or Semantic DataStore
 5:         $\mathbb{B}_{j,AI} \leftarrow$ EstimateDistribution($\mathcal{D}_{AI}, j, k$)
 6:         $\mathbb{B} \leftarrow \mathbb{B} \cup \{\mathbb{B}_{j,H}, \mathbb{B}_{j,AI}\}$
 7:     **end for**
 8:     **return** $\mathbb{B}$
 9: **end procedure**

10: **function** EXTRACTFEATUREVECTOR($X, \mathbb{B}, \mathcal{J}$)
11:     $\mathbf{Z} \leftarrow []$                                                                         ▷ Vector for aggregated log-likelihood ratios
12:     **for** each linguistic level $j \in \mathcal{J}$ **do**
13:         $\mathbf{F}_j \leftarrow \phi_j(X)$                                               ▷ Transform text X to feature sequence for level $j$
14:         $n \leftarrow |\mathbf{F}_j|$
15:         $LLR_j \leftarrow []$                                                                   ▷ Sequence of Log-Likelihood Ratios (LLRs)
16:         **for** $i = k \rightarrow n$ **do**
17:             $c_i \leftarrow (f_{i-k}, \ldots, f_{i-1})$                                         ▷ Context for feature $f_i$
18:             $p_H \leftarrow P(f_i|c_i, Y = H, \mathbb{B}_{j,H})$
19:             $p_{AI} \leftarrow P(f_i|c_i, Y = AI, \mathbb{B}_{j,AI})$
20:             Append $\log(p_{AI}) - \log(p_H)$ to $LLR_j$
21:         **end for**
22:         $z_j \leftarrow \text{mean}(LLR_j)$                                               ▷ Calculate the final feature for level $j$
23:         $\mathbf{Z} \leftarrow \mathbf{Z} \oplus z_j$                                         ▷ Concatenate to the final feature vector
24:     **end for**
25:     **return** $\mathbf{Z}$
26: **end function**

27: **procedure** TRAINANDPREDICT($X_{\text{new}}, \mathcal{D}_{\text{train}}, \mathcal{Y}_{\text{train}}$)
28:     $\mathbb{B} \leftarrow$ BuildDatabases($\mathcal{D}_{\text{train},H}, \mathcal{D}_{\text{train},AI}, \mathcal{J}, k$)
29:     $\mathcal{X}_{\text{features}} \leftarrow \{\text{ExtractFeatureVector}(X, \mathbb{B}, \mathcal{J}) \mid X \in \mathcal{D}_{\text{train}}\}$
30:     $\mathcal{M}_\theta \leftarrow$ XGBoost.fit($\mathcal{X}_{\text{features}}, \mathcal{Y}_{\text{train}}$)
31:
32:     $\mathbf{Z}_{\text{new}} \leftarrow$ ExtractFeatureVector($X_{\text{new}}, \mathbb{B}, \mathcal{J}$)
33:     $P(Y = \text{AI}|\mathbf{Z}_{\text{new}}) \leftarrow \sigma(\mathcal{M}_\theta(\mathbf{Z}_{\text{new}}))$                     ▷ Predict probability for new text
34:     **return** $P(Y = \text{AI}|\mathbf{Z}_{\text{new}})$
35: **end procedure**

---

trials, as summarized in Table 18. All experiments were conducted on a single NVIDIA A800 GPU (80GB).

Table 17: Key hyperparameters for HLD's feature streams.

| Hierarchical Level | Hyperparameters & Components |
|---|---|
| Word | $k = 5$; Tokenizer: `all-MiniLM-L6-v2` |
| Syntactic | $k = 5$; Parser: `spaCy (en_core_web_sm)` |
| Semantic | Context: 4 words, Target: 2 words
Neighbors (M): 4; Encoder: `all-MiniLM-L6-v2`
Indexing: `faiss-gpu` with `nprobe=8` |

Table 18: XGBoost classifier hyperparameter tuning configuration.

| Hyperparameter | Search Space |
|---|---|
| n_estimators | Integer in $[400, 1200]$ |
| learning_rate | Log-uniform in $[10^{-3}, 0.15]$ |
| max_depth | Integer in $[4, 10]$ |
| min_child_weight | Log-uniform in $[0.1, 8.0]$ |
| subsample | Uniform in $[0.6, 1.0]$ |
| colsample_bytree | Uniform in $[0.6, 1.0]$ |

Table 19: Adversarial Attack Methods in the DetectRL Benchmark

| Attack Category | Specific Method | Description |
|---|---|---|
| **Paraphrase Attacks** | DIPPER Paraphraser (Krishna et al., 2023) | Rewrites text using an advanced paraphrasing model to alter phrasing while preserving meaning. |
| | Back-translation | Translates text to another language and back to English (e.g., using an API) to change sentence structure. |
| | Polishing using LLMs | Uses a second LLM to refine or "polish" the initial LLM-generated text, simulating a common editing process. |
| **Perturbation Attacks** | Character-level (DeepWordBug) | Introduces small, adversarial typos and misspellings into the text. |
| | Word-level (TextFooler) | Replaces key words with synonyms that are semantically similar but can confuse detectors. |
| | Sentence-level (TextBugger) | Implements subtle modifications at the sentence level to disrupt linguistic patterns. |
| **Data Mixing** | Multi-LLM Mixing | Creates a single document by combining sentences generated from multiple different LLMs. |
| | LLM-Centered Mixing | Simulates AI-assisted writing by replacing a quarter of the sentences in an LLM-generated text with sentences from a human-written text. |

## C.2 DETAILS OF DATA

The DetectRL(Wu et al., 2024) benchmark is constructed using a combination of human-written texts from high-risk domains and texts generated by powerful, commonly-used Large Language Models (LLMs).

**Human-Written Text.** To ensure real-world relevance, the human-written texts were collected from four distinct domains where LLMs are prone to be misused. A critical aspect of the data collection process was to avoid potential contamination from LLM-generated content. Therefore, all selected human-written data was published before the public release of ChatGPT. The domains are: (i) Academic Writing: Scientific abstracts were sourced from the arXiv Archive (specifically, data from 2002–2017). (ii) News Writing: News articles were taken from the XSum dataset. (iii) Creative Writing: Fictional stories were collected from the Writing Prompts dataset. (iiii) Social Media: User reviews were sourced from the Yelp Reviews dataset.

**LLM-Generated Text.** To create text that closely resembles what is found in real-world applications, the benchmark employed four powerful and widely-used LLMs for text generation:GPT-3.5-turbo, PaLM-2-bison, Claude-instant, Llama-2-70b. The text generation was performed through interactive chat sessions with each model. The prompts were designed to align with the human-written data from each domain, such as providing a paper title to generate an academic abstract or using the first sentence of a human review to have the LLM continue the text.

**Adversarial Attack Methods.** To move beyond simple classification and assess the true resilience of detectors, DetectRL incorporates a multi-faceted adversarial attack framework. These attacks simulate the real-world tactics that a malicious or privacy-conscious user might employ to evade detection. The framework is designed to probe for weaknesses across a spectrum of potential vul-

nerabilities, from prompt-level manipulation to post-generation text alterations. Table 19 provides a detailed summary of the adversarial methods used to challenge the detectors.

### C.3 USE OF LARGE LANGUAGE MODELS

We employed large language models (LLMs) solely as general-purpose writing assistants to refine the manuscript. Specifically, LLMs were used to correct grammar, enhance clarity, and improve phrasing in certain sentences. The models did not contribute to the research design, problem formulation, method development, experimentation, analysis, or overall scientific contributions. Their role was strictly limited to surface-level editing and presentation improvements of the paper.

