# OpenReview forum: "HLD: Approximate Hierarchical Linguistic Distribution Modeling for LLM-Generated Text Detection"
_ICLR.cc/2026/Conference — ICLR 2026 Poster_

### Official Review · Reviewer_DEUM · 2025-10-27

**Soundness:** 3
**Presentation:** 4
**Contribution:** 3
**Rating:** 8
**Confidence:** 4

**Summary:**

The authors propose a novel and effective AI text detection method that models the differences between "Human-Written Text (HWT)" and "Machine-Generated Text (MGT)" as hierarchical linguistic distributions across layers (lexical/morphological → syntactic → semantic). At each layer, they compute features using approximate n-grams, and finally use XGBoost to fuse the four-dimensional features for binary classification.

**Strengths:**

1. Excellent motivation: Zero-shot detection mostly relies on proxy models' token probabilities, which is fragile and computationally expensive; while purely supervised classifiers are accurate, they suffer from weak interpretability and poor generalization. Therefore, the authors cleverly combine perspectives from both approaches.
2. Comprehensive and detailed method comparison
3. Does not rely on source model probabilities, avoiding API access and proxy mismatch issues
4. Provides time efficiency analysis at the end, which is essential in realistic development

**Weaknesses:**

1. Limited LLMs in comparison, and the GPT-3.5 model is already outdated - should add newer models. I understand that adding new models might require extensive experiments, that's fine - the authors could just conduct small-scale experiments to demonstrate effectiveness would be fine.
2. In Table 1, the authors distinguish between zero-shot and supervised detectors. Table 2 and subsequent tables should also make this distinction, for example, using an asterisk to denote zero-shot methods, etc.

**Questions:**

How does the method perform on more recent LLMs like GPT-5?

How does the method handle mixed text (partially human-written, partially AI-generated)?

---

> ### Author Response · Authors · 2025-11-22
> **Response to Weaknesses and Question**
>
> We are grateful for your thorough assessment. To answer the questions you listed, we offer detailed responses below.
>
> >**W1&Q1:** Limited LLMs in comparison, and the GPT-3.5 model is already outdated - should add newer models. I understand that adding new models might require extensive experiments, that's fine - the authors could just conduct small-scale experiments to demonstrate effectiveness would be fine.
> How does the method perform on more recent LLMs like GPT-5?
>
> **Reply:** To further validate our method, we first use multi-new LLMs to generate AI content following the DetectRL benchmark. We then conduct two evaluations: (1) training and testing all training-based methods directly on these newly generated datasets, and (2) training the methods on the original GPT-3.5 data and testing them on the new datasets. The results of these two settings are presented in the tables below.  It indicates that HLD remains effective on these newly emerging models.
>
> |                | DeepSeek-R1 | GPT-5   | Claude 3.5 | Qwen3-Max | GPT-4   | DeepSeek-V3 |
> | :--------------------- | :---------: | :-----: | :--------: | :-------: | :-----: | :---------: |
> | Fast-DetectGPT (ICLR 2024)⁎ |    59.28    | 63.32   |    61.65    |   75.39   | 63.74   |    66.75    |
> | Binoculars (ICML 2024)⁎     |    82.13    | 87.63   |    89.10    |   96.75   | 92.78   |    92.09    |
> | Lastde++ (ICLR 2025)⁎        |    65.57    | 83.46   |    88.40    |   94.18   | 88.55   |    86.88    |
> | RAIDAR (ICLR 2024)          |    93.79    | 88.96   |    90.64    |   90.03   | 91.25   |    92.54    |
> | DPIC (NeurIPS 2024)         |    99.22    | 98.78   |    97.03    |   98.30   | **99.54**   |    99.52    |
> | RoBERTa-base                |    99.52    | **99.61**   |    98.97    |   99.06   | 98.17   |    98.98    |
> | **HLD (Ours)**              |  **99.63**  | 99.54   |  **99.48**  |  **99.43** | 99.14   |  **99.67**  |
>
> |         | DeepSeek-R1 |  GPT-5   | Claude 3.5 | Qwen3-max |  GPT-4   | DeepSeek-V3 |
> | :----------------- | :---------: | :------: | :--------: | :-------: | :------: | :---------: |
> | RAIDAR (ICLR2024)  |    86.06    |  86.61   |   91.44    |   87.26   |  90.87   |    90.35    |
> | DPIC (NeurIPS2024) |    83.65    |  90.70   |   96.85    |   92.95   |  96.64   |    **90.97**    |
> | Robert-base        |    77.04    |  88.64   |   96.89    |   91.38   |  96.42   |    88.60    |
> | **HLD (ours)**     |  **87.46**  | **92.35** |  **97.07** | **94.25** | **96.88** |  89.04  |
>
> >**W2:** In Table 1, the authors distinguish between zero-shot and supervised detectors. Table 2 and subsequent tables should also make this distinction, for example, using an asterisk to denote zero-shot methods, etc.
>
> **Reply:** This is a good suggestion. Following your advice, we have updated all relevant tables to use asterisks (*) to distinguish zero-shot methods from supervised methods.
>
> >**Q2:** How does the method handle mixed text (partially human-written, partially AI-generated)?
>
> **Reply:** Our HLD supports two strategies for handling mixed text. The first strategy is to feed the mixed text directly into the HLD detector. We have run this experiment on DetectRL benchmark by replacing a quarter of the sentences in an LLM-generated text with sentences from a human-written text. As shown in Table 3 Column "Data Mixing'' of the paper, HLD still achives the highest F1-score. The second strategy is to split the input text into sentences or fixed-length segments, compute linguistic feature differences for each segment independently, and use a classifier to determine whether a segment is human-written or AI-generated. Since HLD does not require the entire context, segment-level detection does not degrade detector's performance. As shown in Figure 5(b) of the paper, HLD achieves strong detection results even on short texts. In addition, our method supports token-level heatmaps that precisely pinpoint which segments are more likely AI-generated and which are more likely human-written. This provides an intuitive way for users to understand HLD’s decision process and enhances interpretability.

---

> > ### Comment · Reviewer_DEUM · 2025-11-26
> >
> > I appreciate the authors' rebuttal efforts, especially using different training dataset to train the detector, which can be helpful to identify the in-model and cross-model performance.
> > By the way, is the result based on in-domain, which means the training and testing datatset are from the same domain, like news, writing, or something else?

---

> > > ### Author Response · Authors · 2025-11-27
> > > **Response to Reviewer DEUM**
> > >
> > > Thank you again for your positive feedback on our rebuttal. Your understanding is correct: the result follows an in-domain setting, meaning the training and test sets are drawn from the same domains (arXiv, XSum, WritingPrompts, and Yelp Reviews).

---

### Official Review · Reviewer_xXfd · 2025-10-31

**Soundness:** 3
**Presentation:** 3
**Contribution:** 3
**Rating:** 6
**Confidence:** 3

**Summary:**

The author proposes HLD-Detector: the novel hierarchical linguistic distribution comparison framework that explicitly models differences across three levels—word, syntax, and semantics—using Bayesian likelihood ratio. This framework simultaneously improves detection accuracy, robustness, interpretability, and efficiency.

**Strengths:**

The authors systematically and fully support his hypothesis through comprehensive experiments:
HLD-Detector leads across all benchmarks for DetectRL, including multi-LM and multi-domain.
The SOTA performance of cross-domain/cross-model and attack resistance, validates its robustness; Significantly reduced performance demonstrates the contribution of linguistic feature at each level of its design; In addition, multiple heatmaps allow direct viewing of AI trace phrases/structures.

**Weaknesses:**

1. With the evolution of generative models, the effectiveness of HLD has not been explored when LLM commonly adopts non-maximum likelihood sampling strategies (such as models after RL).

2. While HLD offers advantages in accuracy and interpretability, there is still room for improvement in computational efficiency compared to RoBERTa-base.

**Questions:**

1.Based on the experimental results in Table 3, HLD still appears to be "vulnerable" to domain drift.  Are there any online/offline mechanisms that can improve its robustness? Have you explored which of the three levels of linguistic features might be most affected by domain shift?

2.Some more implementation details of HLD are missing, like the hyperparameters smoothing factor $\delta$ , $\tau_{ctx}$ , etc.

---

> ### Author Response · Authors · 2025-11-22
> **Response to Weaknesses1 & 2**
>
> Thank you for the detailed insights and suggestions, which helped us improve the clarity and completeness of the paper. In response to the issues you raised, we provide point-by-point clarifications as follows.
>
> > **W1:** With the evolution of generative models, the effectiveness of HLD has not been explored when LLM commonly adopts non-maximum likelihood sampling strategies (such as models after RL).
>
> **Reply:** To address this concern, we add a new experiment on DeepSeek-R1 and Qwen3-Max, two representative models that are pre-trained after RL. We follow the setting of DetectRL and use Deepseek-R1 and Qwen3-Max to generate the samples. The following two tables report the corresponding results. In the first table, all training-based methods are trained and evaluated on DeepSeek-R1 and Qwen3-Max data. In the second table, all training-based methods are trained on GPT-3.5 and evaluated on DeepSeek-R1 and Qwen3-Max. From the two tables, we can observe that HLD remains highly effective on LLM pre-trained after RL.
>
> | | DeepSeek-R1 (AUC) | DeepSeek-R1 (F1) | Qwen3-Max (AUC) | Qwen3-Max (F1) |
> | :--- | :---: | :---: | :---: | :---: |
> | Fast-DetectGPT(ICLR 2024) | 59.28 | 49.43 | 75.39 | 64.23 |
> | Binoculars(ICML2024) | 82.13 | 59.50 | 96.75 | 91.14 |
> | Lastde++(ICLR2025) | 65.57 | 68.70 | 94.18 | 86.47 |
> | RAIDAR(ICLR2024) | 93.79 | 87.21 | 90.03 | 81.64 |
> | DPIC(NeurIPS2024) | 99.22 | 95.90 | 98.30 | 92.98 |
> | Robert-base | 99.52 | 96.52 | 99.06 | 95.49 |
> | **HLD(ours)** | **99.63** | **97.87** | **99.43** | **97.29** |
>
> | | DeepSeek-R1 (AUC) | DeepSeek-R1 (F1) | Qwen3-Max (AUC) | Qwen3-Max (F1) |
> | :--- | :---: | :---: | :---: | :---: |
> | RAIDAR(ICLR2024) | 86.06 | 80.56 | 87.26 | 81.16 |
> | DPIC(NeurIPS2024) | 83.65 | 80.82 | 92.95 | 87.79 |
> | Robert-base | 77.04 | 68.22 | 91.38 | 86.23 |
> | **HLD(ours)** | **87.46** | **84.51** | **94.25** | **90.41** |
>
> >**W2:** While HLD offers advantages in accuracy and interpretability, there is still room for improvement in computational efficiency compared to RoBERTa-base.
>
> **Reply:** Compared to a RoBERTa-base classifier, HLD is slightly slower during inference because it requires generating semantic feature sequences using pretrained models for the input texts. However, its inference efficiency is still significantly higher than those zero-shot methods which rely on large proxy models. In future work, we plan to further optimize the extraction of semantic features to continuously improve HLD’s runtime efficiency.

---

> ### Author Response · Authors · 2025-11-22
> **Response to Question1 & 2**
>
> >**Q1:** Based on the experimental results in Table 3, HLD still appears to be "vulnerable" to domain drift. Are there any online/offline mechanisms that can improve its robustness? Have you explored which of the three levels of linguistic features might be most affected by domain shift?
>
> **Reply:** For the question on vulnerability to domain shift, we agree that HLD exhibits a slightly larger performance gap between the in-domain arXiv data and the out-of-domain Writing data. Despite this, HLD consistently outperforms all baseline methods in out-of-domain settings, and on several out-of-domain datasets it achieves around 90 AUROC scores. To further improve cross-domain robustness, we believe that expanding the feature library size would be beneficial. We currently build the feature library from only 1,800 human texts and 1,800 AI-generated texts, which is relatively small and can be easily enlarged in future work.
>
> For the analysis of linguistic features, we additionally conduct an experiment using single-level feature. As shown in the following table, shallow word-level features exhibit the largest performance fluctuation. They achieve the highest AUROC (97.88) on the in-domain Arxiv task but drop to 73.48 in the out-of-domain Writing data. This suggests that in highly formulaic domains such as arXiv, word-level feature effectively reflect differences between human and AI text. While, in the stylistically flexible Writing domain, word-level feature is more easily disrupted, leading to weaker performance. This motivates us to leverage hierarchical linguistic features to comprehensively capture differences between human and machine-generated text in multi aspects, enabling robust cross-domain detection.
>
> |    | Arxiv (AUC) | Arxiv (F1) | Xsum (AUC) | Xsum (F1) | Writing (AUC) | Writing (F1) | Review (AUC) | Review (F1) |
> | :------------- | :---------: | :---------: | :---------: | :---------: | :-----------: | :-----------: | :-----------: | :-----------: |
> | Word           |    97.88    |    95.44    |    80.09    |    72.89    |     73.48     |     65.71     |     87.31     |     80.03     |
> | POS            |    97.75    |    92.60    |    82.38    |    74.63    |     76.89     |     63.96     |     83.86     |     75.08     |
> | Dep            |    96.37    |    91.50    |    79.68    |    74.27    |     77.12     |     70.55     |     80.72     |     73.33     |
> | Semantic       |    92.26    |    87.96    |    80.82    |    72.21    |     76.58     |     70.91     |     78.60     |     72.08     |
>
> >**Q2:** While HLD offers advantages in accuracy and interpretability, there is still room for improvement in computational efficiency compared to RoBERTa-base.
>
> **Reply:** We thank the reviewer for pointing this out. The hyperparameters are set as $\delta = 1 \times 10^{-6}$  and $\tau_{\text{ctx}} = 0.1$ and we have updated these hyperparameter settings in our paper.

---

### Official Review · Reviewer_KXqA · 2025-11-02

**Soundness:** 3
**Presentation:** 2
**Contribution:** 3
**Rating:** 6
**Confidence:** 4

**Summary:**

This paper proposes a new model for detecting machine generated texts, called HLD-detector. HLD-detector uses word-level differences, syntactic-level differences, and semantic embeddings as its features, and adopts kNN database for estimate density of those features. With XGBoost classifier, the authors claimed that their model achieved highest performance in multi-LLM setting and multi-domain setting. Also, they claimed HLD-detector showed robust performance on cross-domain, cross-model, and diverse attack scenarios. An ablation study showed that those proposed features work well in detecting MGTs. In addition to these experiments, the authors suggested other experimental results such as interpretability, impact of n-gram or text length, impact of data scale, Chinese MGTs, and efficiency of their methods, to emphasize their contribution.

**Strengths:**

- Kernel density estimation methods for MGT detection, using a simple component of a text, n-gram.
- Multi-dimensional experiments that reveals the actual applicability of HLD-detector.

**Weaknesses:**

- Though the authors attempted to follow DetectRL benchmark, the source models seem outdated. Because recent models learned more human-like wordings and phrases, recently researchers thought that the syntactic-level features might not suitable to detect MGTs. Refer to [1] and [2]. Also see Question A.

[1] Z. Kim, et al. Threads of Subtlety: Detecting Machine-Generated Texts Through Discourse Motifs. arXiv preprint arXiv:2402.10586 (2024).
[2] H. Park et al., DART : An AIGT Detector using AMR of Rephrased Text, NAACL 2025

**Questions:**

## Question A. Source models

A1. The naturalness of text generated by source models might affect the result. Recent advances in language modeling proceeds to generate more human-like texts, and this continuously introduces challenges in MGT detection. Although DetectRL can reveal the performance and possibility of detecting MGTs, a reader might ask whether these models can be successfully applicable to recent models. I understand that the authors tried to claim that their model could be applied to recent models with cross-model tests, I recommend adding one more experiments showing whether the proposed model could be successfully detect GPT-4 or Claude 3.5, even if they were trained on previous version of those families.
A1-1. In the same vein, the table might show a kind of ceiling effect: All performance scores were near 100. What if we ran the same experiment on more complex or difficult datasets?

A2. Usually, we don't know source models exactly. When discriminating MGTs from HWTs, a user could not know which source model is a possible source of given text. So, models like RAIDAR or [2] attempted to expand the problem into multi-class classification problem instead of a binary-class problem. Could this model be extended to those cases easily?

## Question B. N-grams and semantics.

B1. As a researcher also working on semantic representations, claiming "N-grams" have "high-level semantics" seems inappropriate. This is because N-gram and its semantic embedding only mirror its pragmatic context, rather than the deep semantics (such as events or participants), and N-gram often cannot model long-distance dependencies. And, this is why [1] and [2] attempted to adopt higher-level of semantic representations in detecting MGTs. Why didn't the authors consider such high-level semantic representations?

B2. Authors noted that the designed features contributed to the performance, in Table 4. But the score seems not much differ, especially on XSum. This makes me raise two following questions.
B2-1. Are the differences statistically significant? Could we claim that the difference was not produced by chance (i.e., Type I error)?
B2-2. Why do the features behave differently across domains? Do the authors have any other explanations on that? Answering this might provide some valuable insights to the readers also.

---

> ### Author Response · Authors · 2025-11-22
> **Response to Weaknesses & Question A1**
>
> We appreciate your careful evaluation of our work and the valuable comments you provided. We address your comments and questions one by one in the following response.
>
> >**W1:** Though the authors attempted to follow DetectRL benchmark, the source models seem outdated. Because recent models learned more human-like wordings and phrases, recently researchers thought that the syntactic-level features might not suitable to detect MGTs. Refer to [1] and [2]. Also see Question A.
>
> **Reply:** We understand the concern that “syntactic features" may become less effective in recent years. We know that in certain domains, syntactic features contribute relatively little to AIGC detection. As shown in Table 4, in domains such as arXiv and review, syntactic structures are highly standardized, so their contribution is limited. In contrast, in domains such as xsum and writing, syntactic features become an important signal for distinguishing human-written from machine-generated text (MGT). This observation suggests that, although LLMs generate more human-like content, syntactic features still be useful to discriminate MGT in specific tasks. KatFishNet[1] (ACL'25) also reports notable differences between humans and AI at the POS level in certain domains.
> Based on these observations, we conclude that syntactic features remain an important factor for distinguishing MGT. However, relying solely on a single syntactic level feature is insufficient for building a robust detector. To address this, we also leverages lexical and semantic-level feature, allowing complementary information across diverse domains and enabling HLD to maintain robust detection performance.
>
> >**QA1:** The naturalness of text generated by source models might affect the result. Recent advances in language modeling proceeds to generate more human-like texts, and this continuously introduces challenges in MGT detection. Although DetectRL can reveal the performance and possibility of detecting MGTs, a reader might ask whether these models can be successfully applicable to recent models. I understand that the authors tried to claim that their model could be applied to recent models with cross-model tests, I recommend adding one more experiments showing whether the proposed model could be successfully detect GPT-4 or Claude 3.5, even if they were trained on previous version of those families.
>
> **Reply:** We additionally evaluate our detector that trained exclusively on GPT-3.5 on the latest LLMs, including Qwen3, DeepSeek-V3, DeepSeek-R1, GPT-4o, GPT-5, and Claude 3.5. The experimental results are summarized below:
>
> |         | DeepSeek-R1 |  GPT-5   | Claude 3.5 | Qwen3-max |  GPT-4   | DeepSeek-V3 |
> | :----------------- | :---------: | :------: | :--------: | :-------: | :------: | :---------: |
> | RAIDAR (ICLR2024)  |    86.06    |  86.61   |   91.44    |   87.26   |  90.87   |    90.35    |
> | DPIC (NeurIPS2024) |    83.65    |  90.70   |   96.85    |   92.95   |  96.64   |    **90.97**    |
> | Robert-base        |    77.04    |  88.64   |   96.89    |   91.38   |  96.42   |    88.60    |
> | **HLD (ours)**     |  **87.46**  | **92.35** |  **97.07** | **94.25** | **96.88** |  89.04  |
>
> The evaluation shows that our HLD detector continues to maintain a leading position even on the generation dataset of latest language models.
>
> [1] KatFishNet: Detecting LLM-Generated Korean Text through Linguistic Feature Analysis

---

> ### Author Response · Authors · 2025-11-22
> **Response to Question A1-1 & A2**
>
> > **QA1-1:** In the same vein, the table might show a kind of ceiling effect: All performance scores were near 100. What if we ran the same experiment on more complex or difficult datasets?
>
> **Reply:** In the in-domain experiments (Table 1), several supervised models indeed reach near-perfect performance. However, DetectRL emphasizes that cross-domain evaluation is the more realistic and challenging scenario. Following this, we evaluate both the baselines and HLD under cross-domain shifts (results are shown in Paper's table 2). To further increase the difficulty, we additionally test generalization to newest LLMs (results are shown in the previous question), and we directly apply the HLD model trained on DetectRL to evaluate its performance on the complex benchmark MIRAGE[1] (ACM MM'25) (results are shown in the following table). Across all these experiments, we observe that supervised baselines that achieve superior in-domain performance suffer substantial drops when facing distribution shifts. In contrast, our method exhibits a smaller performance drop under out-of-domain settings, demonstrating the better generalization compared to other approaches.
>
> |                | MIRAGE-DIG（Auc） | MIRAGE-DIG（TPR@5%） | MIRAGE-SIG（Auc） | MIRAGE-SIG（TPR@5%） |
> | :---------------------- | :---------------: | :------------------: | :---------------: | :------------------: |
> | DNA-GPT(ICLR 2024)*      |       57.33       |         7.76         |       57.59       |         8.13         |
> | Fast-DetectGPT(ICLR 2024)* |       77.68       |        43.10         |       77.06       |         42.00        |
> | ImBD(AAAI 2025 oral)     |       85.97       |        40.65         |       86.12       |         41.83        |
> | DetectAnyLLM(MM2025)     |       95.25       |        77.70         |       95.26       |         77.22        |
> | **HLD(ours)**            |       **96.72**       |        **86.64**         |       **96.95**       |         **86.80**        |
>
> > **QA2:** Usually, we don't know source models exactly. When discriminating MGTs from HWTs, a user could not know which source model is a possible source of given text. So, models like RAIDAR or [2] attempted to expand the problem into multi-class classification problem instead of a binary-class problem. Could this model be extended to those cases easily?
>
> **Reply:** We thank the reviewer for the suggestion. We implement a simple three-category classification pipeline based on the HLD framework on a detection dataset containing human-written text, and content generated by DeepSeek-R1 and GPT-5. Specifically, we constructed two binary HLD detectors: (i) an AI vs. Human binary classifier to distinguish between AI-generated and human-written text; and (ii) a DeepSeek vs. GPT binary classifier to identify the specific AI source. At inference time, a given text is first evaluated by AI vs. Human detector. If it is predicted as AI-generated, we further feed the text into the DeepSeek vs. GPT binary model for source identification. The experimental results (shown below) demonstrate that our HLD-based pipeline achieves accurate three-category classification. In future work, we plan to extend this approach to support more categories of text sources.
>
> | | | Human | GPT-5 | DeepSeek-R1 |
> | :--- | :---: | :---: | :---: | :---: |
> | **Precision** | Robert-base | **99.26** | 79.21 | 77.98 |
> | | **HLD(ours)** | 99.21 | **93.93** | **95.28** |
> | **Recall** | Robert-base | **100** | 76.20 | 78.60 |
> | | **HLD(ours)** | **100** | **94.40** | **92.90** |
>
> [1] DetectAnyLLM: Towards Generalizable and Robust Detection of Machine-Generated Text Across Domains and Models

---

> ### Author Response · Authors · 2025-11-22
> **Response to Question B1 & B2**
>
> > **QB1:** As a researcher also working on semantic representations, claiming "N-grams" have "high-level semantics" seems inappropriate. This is because N-gram and its semantic embedding only mirror its pragmatic context, rather than the deep semantics (such as events or participants), and N-gram often cannot model long-distance dependencies. And, this is why [1] and [2] attempted to adopt higher-level of semantic representations in detecting MGTs. Why didn't the authors consider such high-level semantic representations?
>
> **Reply:** We thank the reviewer for the insightful comments and agree that N-grams do not capture high-level semantics. In our work, N-grams are only used to approximate semantic feature distributions, while the actual semantic representations are obtained via the semantic model SBERT. We will clarify this point more clearly in the revised paper to avoid potential misunderstandings.
>
> With respect to the concern that N-grams cannot model long-distance dependencies, this is indeed a limitation of N-grams. Fortunately, our framework relies on compairing distributional differences between AI-generated and human text,  and short-term distributions modeling is sufficient to capture the difference across two distributions.
>
> Regarding the suggestion to incorporate higher-level AMR representations, we agree that AMR is a promising high-level feature for AIGC detection. However, the extraction of AMR features necessitates additional pretrained AMR models, which would further increase the detector’s runtime. In future work, we would incorporate AMR features into our HLD framwork and conduct more in-depth comparative experiments.
>
> > **QB2:** Authors noted that the designed features contributed to the performance, in Table 4. But the score seems not much differ, especially on XSum. This makes me raise two following questions. B2-1. Are the differences statistically significant? Could we claim that the difference was not produced by chance (i.e., Type I error)? B2-2. Why do the features behave differently across domains? Do the authors have any other explanations on that? Answering this might provide some valuable insights to the readers also.
>
> **Reply: B2-1:** We additionally repeat the ablation experiments five times on the XSum dataset and report the average results across these runs. We also compute the p-values for the performance differences between the full HLD model and the variants with specific features removed on the Xsum. The summarized results are shown below. The p value is much smaller than 0.05, therefore the difference is statistically significant.
>
> |                |     AUROC      | p-value (vs. all) |
> | :------------: | :-------------: | :---------------: |
> | HLD (all)       |   89.07 ± 0.39  |         --       |
> | -word          |   87.80 ± 0.33  |     4.60e-03      |
> | -POS           |   85.58 ± 0.25  |     4.03e-06      |
> | -Dep           |   86.79 ± 0.35  |     6.37e-04      |
> | -Semantic      |   88.21 ± 0.29  |     3.75e-04      |
>
> **B2-2:** We believe that the differences in the contributions of linguistic features across domains can be explained by domain-specific writing conventions and text characteristics. For highly structured texts such as ArXiv abstracts or Yelp reviews, due to LLMs’ tendency to frequently employ fixed academic expressions,, it allows shallow word-level features to easily distinguish between human- and AI-generated text.
>
> In contrast, the flexible vocabulary choices and diverse word-combination patterns characteristic of stories make word-level features insufficient for reliably distinguishing AI-generated text in the story-writing domain. We observe that story-generation tasks emphasize narrative coherence and plot development. Consequently, linguistic features at the semantic and syntactic levels are more effective in capturing differences in coherence and fluency between human- and machine-generated content, thereby enabling more accurate classification.
>
> For fact-focused XSum news texts, the discriminative signals between human and AI-generated text are distributed across multiple linguistic levels, including the word, syntactic, and semantic levels, with no single level of feature dominating. As a result, the contributions of different features to detection performance are relatively balanced.

---

### Official Review · Reviewer_jiri · 2025-11-13

**Soundness:** 3
**Presentation:** 3
**Contribution:** 3
**Rating:** 2
**Confidence:** 4

**Summary:**

This paper introduces HLD-Detector, a hierarchical linguistic distribution framework for detecting machine-generated text. Instead of relying on proxy-model probabilities or opaque neural classifiers, the method explicitly models word-level, syntactic, and semantic distribution differences using n-gram statistics and a KDE-based semantic estimator, then combines them via an XGBoost classifier. Experiments on DetectRL show strong performance across multi-LLM, multi-domain, transfer, and adversarial settings, with notable efficiency and interpretability advantages. Ablations confirm the complementary roles of each linguistic layer.

**Strengths:**

The paper has a good motivation, whose hybrid design combining interpretable distribution modeling with supervised fusion, bridging the gap between zero-shot methods and black-box supervised classifiers.

The paper is also easy to follow.

**Weaknesses:**

1. Although the authors cited top-tier recent LLMs like DeepSeek-r1, unfortunately empirical evaluations cover early LLMs like Claude,PaLM-2, and LLaMA-2. As a paper in late 2025, I think it is necessary for an AI text detector to include latest LLMs, including: Qwen3, DeepSeek-V3/R1, GPT4, et cetera.

2. Besides, I am curious why Roberta-base, the simplest baseline, outperforms nearly all other methods in Table 3. Is that a fair comparison? I guess the reason is that the training corpus that is used for training Roberta is a strong one, hinting potential unfairness.

3. Unfortunately, the method is confined to English. I think more languages could be included to improve the generality of the proposed method.

**Questions:**

Now commercial detectors are widely used, is the proposed HLD method comparable to these detectors, e.g. GPT-Zero and ZeroGPT?

---

> ### Author Response · Authors · 2025-11-22
> **Response to Weaknesses1**
>
> Thank you for your thoughtful review and constructive feedback. Regarding your questions, we address each of them in detail below.
>
> >**W1:** Although the authors cited top-tier recent LLMs like DeepSeek-r1, unfortunately empirical evaluations cover early LLMs like Claude,PaLM-2, and LLaMA-2. As a paper in late 2025, I think it is necessary for an AI text detector to include latest LLMs, including: Qwen3, DeepSeek-V3/R1, GPT4, et cetera.
>
> **Reply:** We acknowledge the reviewer’s concern regarding the necessity of evaluating detectors on recent LLMs. In the updated experiments, we additionally evaluate our detectors on the latest large language models, including Qwen3, DeepSeek-V3, DeepSeek-R1, GPT-4, GPT-5 and Claude 3.5, ensuring the experiments reflect the performance of detectors on the latest LLMs.
> Specifically, following the DetectRL (NeurIPS'24) setting, we get AI-generated texts across ArXiv, XSum, Writing, and Yelp using multiple generation strategies for each of the newly added LLMs. Below, we present the results in AUROC of baseline methods and our HLD detector on these newly generated datasets.
>
> |                | DeepSeek-R1 | GPT-5   | Claude 3.5 | Qwen3-Max | GPT-4   | DeepSeek-V3 |
> | :--------------------- | :---------: | :-----: | :--------: | :-------: | :-----: | :---------: |
> | Fast-DetectGPT (ICLR 2024)⁎ |    59.28    | 63.32   |    61.65    |   75.39   | 63.74   |    66.75    |
> | Binoculars (ICML 2024)⁎     |    82.13    | 87.63   |    89.10    |   96.75   | 92.78   |    92.09    |
> | Lastde++ (ICLR 2025)⁎        |    65.57    | 83.46   |    88.40    |   94.18   | 88.55   |    86.88    |
> | RAIDAR (ICLR 2024)          |    93.79    | 88.96   |    90.64    |   90.03   | 91.25   |    92.54    |
> | DPIC (NeurIPS 2024)         |    99.22    | 98.78   |    97.03    |   98.30   | **99.54**   |    99.52    |
> | RoBERTa-base                |    99.52    | **99.61**   |    98.97    |   99.06   | 98.17   |    98.98    |
> | **HLD (Ours)**              |  **99.63**  | 99.54   |  **99.48**  |  **99.43** | 99.14   |  **99.67**  |
>
> These results show that our HLD detector continues to achieve state-of-the-art performance even on these more advanced models. To further examine the generalization ability of HLD to newly emerged LLMs, we used the HLD trained exclusively on GPT-3.5 and tested it on data generated by these new models. The results (AUROC) are as follows:
>
> |         | DeepSeek-R1 |  GPT-5   | Claude 3.5 | Qwen3-max |  GPT-4   | DeepSeek-V3 |
> | :----------------- | :---------: | :------: | :--------: | :-------: | :------: | :---------: |
> | RAIDAR (ICLR2024)  |    86.06    |  86.61   |   91.44    |   87.26   |  90.87   |    90.35    |
> | DPIC (NeurIPS2024) |    83.65    |  90.70   |   96.85    |   92.95   |  96.64   |    **90.97**    |
> | Robert-base        |    77.04    |  88.64   |   96.89    |   91.38   |  96.42   |    88.60    |
> | **HLD (ours)**     |  **87.46**  | **92.35** |  **97.07** | **94.25** | **96.88** |  89.04  |
>
> As shown, HLD demonstrates strong generalization capabilities, highlighting the practical value of our approach. Additionally, we selected the DetectRL since it includes more challenging cross-domain and adversarial-attack scenarios, allowing a more comprehensive evaluation of different detectors. To further address your concerns, we also adapt the newly released 2025 AI-generated text detection benchmark MIRAGE[1] (ACM MM’25), which covers 5 text domains, 17 advanced LLMs including 13 proprietary ones, and re-evaluated our HLD detector. Notably, we directly used the model trained on DetectRL. As shown in the table below, our detector continues to achieve excellent performance on MIRAGE, further validating the robustness and effectiveness of our framework.
>
> |                | MIRAGE-DIG（Auc） | MIRAGE-DIG（TPR@5%） | MIRAGE-SIG（Auc） | MIRAGE-SIG（TPR@5%） |
> | :---------------------- | :---------------: | :------------------: | :---------------: | :------------------: |
> | DNA-GPT(ICLR 2024)*      |       57.33       |         7.76         |       57.59       |         8.13         |
> | Fast-DetectGPT(ICLR 2024)* |       77.68       |        43.10         |       77.06       |         42.00        |
> | ImBD(AAAI 2025)     |       85.97       |        40.65         |       86.12       |         41.83        |
> | DetectAnyLLM(MM2025)     |       95.25       |        77.70         |       95.26       |         77.22        |
> | **HLD(ours)**            |       **96.72**       |        **86.64**         |       **96.95**       |         **86.80**        |
>
> [1] DetectAnyLLM: Towards Generalizable and Robust Detection of Machine-Generated Text Across Domains and Models

---

> ### Author Response · Authors · 2025-11-22
> **Response to Weaknesses2 & 3 and Question**
>
> >**W2:** Besides, I am curious why Roberta-base, the simplest baseline, outperforms nearly all other methods in Table 3. Is that a fair comparison? I guess the reason is that the training corpus that is used for training Roberta is a strong one, hinting potential unfairness.
>
> **Reply:** For the adversarial experiments, we clarify that all supervised baselines and our method are trained on the same un-attacked Direct dataset. This design ensures a fully controlled and fair comparison across all models.
> Regarding the strong performance of RoBERTa on certain adversarial sets, DetectRL has shown that RoBERTa performs well when training and test data come from the same distribution. Since the Perturbation attack introduces only minor token-level edits, the resulting test distribution remains very close to the training data (Direct). Therefore, RoBERTa’s performance is minimally affected. However, under the more challenging Paraphrase attack, its performance drops sharply. In contrast, our HLD method maintains consistently high AUROC and F1 scores across all attack types, demonstrating its stronger robustness.
> On the concern that RoBERTa’s pretraining corpus may give it an inherent advantage, prior work such as IMBD[1] (AAAI'25) evaluated the pretrained RoBERTa model without any fine-tuning and reported AUROC scores close to 0.5. This near-random performance indicates that the pretraining data contributes little to RoBERTa's ability to detect machine-generated text. The effectiveness of RoBERTa in our experiments therefore arises from fine-tuning stage.
>
> >**W3:** Unfortunately, the method is confined to English. I think more languages could be included to improve the generality of the proposed method.
>
> **Reply:** Our current study primarily focuses on English, which we acknowledge as an important limitation. To explore the effectiveness of our method on non-English data, we have already conducted experiment on a Chinese dataset and reported the results in the “EvaluationonNon-EnglishDataset” section of our paper. To provide a broader assessment of HLD’s detection performance on other non-English datasets, we further conducted experiments on the M4GT-Bench[2] (ACL'24) Russian dataset and KatFishNet[3] (ACL'25). The results are as follows:
>
> |                | Russian (auc) | Russian (f1) | Korean (auc) | Korean (f1) |
> | :---------------------- | :-----------: | :-----------: | :-----------: | :-----------: |
> | Fast-DetectGPT(ICLR 2024)* |     66.65     |     68.79     |     65.44     |     46.21     |
> | Robert                  |     85.29     |     77.73     |     75.98     |     66.19     |
> | **HLD(ours)**            |   **95.84**   |   **91.25**   |   **84.97**   |   **76.36**   |
>
> Our HLD attains the highest AUC and F1 score, validating its effectiveness for multilingual applications.
>
> >**Q1:** Now commercial detectors are widely used, is the proposed HLD method comparable to these detectors, e.g. GPT-Zero and ZeroGPT?
>
> **Reply:** We have already compared the HLD with GPTZero on the MGTBench (CCS'24) and reported the details in paper's Appendix B.2. In addition, we compare our method against the commercial models GPT-Zero and ZeroGPT on RAID[4] (ACL'24) dataset. Due to the large dataset size, we randomly sample 2000 samples from the training set of RAID and the results are presented in Accuracy@FPR=5%. The results demonstrate that our model remains highly competitive with commercial detecotrs.
>
> |        | Chat models(llama-c,mistral-c,mpt-c) |        | Non-chat models(mistral,mpt,gpt2) |        | Chat models(c-gpt,gpt4,cohere) | Non-chat models(cohere,gpt3) |
> | :-------------: | :----------------------------------: | :----: | :---------------------------------: | :----: | :------------------------------: | :------------------------------: |
> | penalty        |                  ×                   |   √    |                  ×                  |   √    |                 ×                  |                 ×                  |
> | gptzero        |                 **98.4**                 |  82.5  |                  9.4                |  4.8   |                **88.5**               |                53.4               |
> | zerogpt        |                 97.7                 |  72.5  |                 16.0                |  0.3   |                65.8               |             **72.7**              |
> | **HLD(ours)**  |               92.5               | **88.9**|               **33.1**              | **46.7**|               83.7             |               64.1             |
>
>
> [1] Imitate Before Detect: Aligning Machine Stylistic Preference for Machine-Revised Text Detection
>
> [2] M4GT-Bench: Evaluation Benchmark for Black-Box Machine-Generated Text Detection
>
> [3] KatFishNet: Detecting LLM-Generated Korean Text through Linguistic Feature Analysis
>
> [4] RAID: A Shared Benchmark for Robust Evaluation of Machine-Generated Text Detectors

---

> > ### Comment · Reviewer_jiri · 2025-11-22
> > **Concerns Addressed**
> >
> > I appreciate the authors' rebuttal efforts. After reading their rebuttal, I think my concerns are mostly addressed. I have raised my score.

---

> > > ### Author Response · Authors · 2025-11-24
> > > **Appreciate**
> > >
> > > Thank you! We are pleased to know that our rebuttal has addressed your concerns, and these experiments will be included in the revised paper.

---

### Author Response · Authors · 2025-12-03
**Summary of Rebuttal Updates and Reviewer Feedback**

Dear Area Chair,

We sincerely appreciate your time and effort in handling our submission. We provide this summary to outline the key rebuttal updates and reviewer interactions from the discussion phase. Updates and new results based on reviewer feedback are included in the revision and highlighted in blue.

**1. Key Rebuttal Updates**

During the discussion phase, we addressed the reviewers' primary concerns by:

*   **Validating on Latest Models (Reviewers jiri, KXqA, xXfd, DEUM):** We extended the evaluation scope to newly released models, specifically DeepSeek-R1, DeepSeek-V3, GPT-4, GPT-5, Qwen3-Max, and Claude 3.5. HLD maintained SOTA performance and outperformed recent baselines (ICLR'24/25, NeurIPS'24, ICML'24). (Detailed in Revision `Appendix B.1` and `Table 11`).
*   **Expanding Benchmarks & Generalization (Reviewers jiri, KXqA):** We verified HLD on the MIRAGE benchmark, achieving highest AUROC and TPR@5%. (Detailed in Revision `Appendix B.2` and `Table 14`); We compared HLD against GPTZero and ZeroGPT, showing that HLD is highly competitive in accuracy. (Detailed in Revision `Appendix B.2` and `Table 15`); We added experiments on Russian and Korean datasets, outperforming RoBERTa by 10-20%. (Detailed in Revision `Section 3.2` and `Figure 6`).
*   **Deepening Methodological Analysis (Reviewers KXqA, DEUM):** We implemented a pipeline for multi-class source attribution (Human vs. GPT-5 vs. DeepSeek-R1) with high classification accuracy. (Detailed in Revision `Appendix B.1` and `Table 12`); We provided p-value analysis ($p < 0.01$) to prove the statistical significance of our hierarchical features (Detailed in Revision `Appendix B.3` and `Table 16`) and demonstrated our strong capabilities in fine-grained mixed text detection.

**2. Received Reviewers' Feedback After Rebuttal**

Crucially, we wish to highlight the specific feedback received during the discussion period, which took place before the recent platform-wide incident:

*   **Reviewer jiri (Score Raised: from 2 → 6, Nov 22):** After reviewing our additional experiments, Reviewer jiri explicitly stated: *"I appreciate the authors' rebuttal efforts. After reading their rebuttal, I think my concerns are mostly addressed. I have raised my score."*
*   **Reviewer DEUM (Nov 26):** Provided positive validation of our new experimental settings, noting: *"I appreciate the authors' rebuttal efforts ... which can be helpful to identify the in-model and cross-model performance."*

We fully respect the program chairs' decision to revert the review scores to ensure fairness across all submissions. This summary is provided solely to assist the Area Chair in capturing the progress developed during the discussion phase.

Best regards,

Submission 15775 Authors

---

### Meta-Review · Area_Chair_ERVG · 2026-01-07

**Summary:**

This paper proposes HLD-Detector, a hierarchical linguistic distribution framework for detecting LLM-generated text by modeling differences across word, syntactic, and semantic levels using n-gram statistics and KDE-based estimation. The method bridges zero-shot approaches and supervised classifiers by combining interpretable distribution modeling with XGBoost fusion. Reviewers acknowledged the solid motivation, comprehensive experiments, and practical efficiency advantages. While initial concerns centered on the use of outdated source models and limited language coverage, the authors provided extensive additional experiments during rebuttal, demonstrating strong performance on latest models (DeepSeek-R1, GPT-5, Claude 3.5, Qwen3-Max), new benchmarks (MIRAGE), commercial detector comparisons, and multilingual datasets (Russian, Korean). The hierarchical design effectively addresses the limitations of single-level detectors, and the method shows robust cross-domain and cross-model generalization. Given the thorough experimental validation and the practical value of an efficient, interpretable detector, this paper makes a meaningful contribution to the field of AI-generated text detection.

**Reviewer Concerns:**

The primary concerns regarding evaluation on outdated LLMs were comprehensively addressed through new experiments on six recent models including DeepSeek-R1, GPT-5, and Claude 3.5, where HLD maintained SOTA performance. The concern about English-only evaluation was mitigated by additional experiments on Russian and Korean datasets showing approximately 10% improvements over RoBERTa. Questions about statistical significance were resolved with p-value analysis (p < 0.01) confirming the contribution of hierarchical features. The concern about comparison with commercial detectors was addressed through experiments against GPTZero and ZeroGPT. The question about multi-class source attribution was answered with a working pipeline demonstrating high classification accuracy. Outstanding concerns include the slightly lower computational efficiency compared to RoBERTa-base, the inherent limitations of n-grams in capturing long-distance dependencies (though the authors argue distributional comparison mitigates this), and some vulnerability to domain drift as evidenced by performance drops from structured domains like arXiv to flexible domains like Writing.

**Reviewer Scores:**

Reviewer jiri explicitly raised their score from 2 to 6 during the discussion period, stating their concerns were "mostly addressed" by the rebuttal experiments. Reviewer KXqA (score 6) received detailed responses addressing source model concerns, multi-class extension, and statistical significance, and would likely maintain or slightly increase their score given the comprehensive experimental additions. Reviewer xXfd (score 6) had concerns about RL-trained models addressed through DeepSeek-R1 and Qwen3-Max experiments, supporting score maintenance. Reviewer DEUM (score 8) engaged positively with the rebuttal, confirming satisfaction with the in-model and cross-model experimental setup, and would likely maintain their supportive score.

---

### Decision · Program_Chairs · 2026-01-26

Accept (Poster)